# Vocabulary-free Image Classification

**Alessandro Conti**[1]     **Enrico Fini**[1]     **Massimiliano Mancini**[1]
**Paolo Rota**[1]     **Yiming Wang**[2]     **Elisa Ricci**[1,2]

[1]University of Trento     [2]Fondazione Bruno Kessler (FBK)

## Abstract

Recent advances in large vision-language models have revolutionized the image classification paradigm. Despite showing impressive zero-shot capabilities, a pre-defined set of categories, *a.k.a.* the vocabulary, is assumed at test time for composing the textual prompts. However, such assumption can be impractical when the semantic context is unknown and evolving. We thus formalize a novel task, termed as Vocabulary-free Image Classification (VIC), where we aim to assign to an input image a class that resides in an unconstrained language-induced semantic space, without the prerequisite of a known vocabulary. VIC is a challenging task as the semantic space is extremely large, containing millions of concepts, with hard-to-discriminate fine-grained categories. In this work, we first empirically verify that representing this semantic space by means of an external vision-language database is the most effective way to obtain semantically relevant content for classifying the image. We then propose Category Search from External Databases (CaSED), a method that exploits a pre-trained vision-language model and an external vision-language database to address VIC in a training-free manner. CaSED first extracts a set of candidate categories from captions retrieved from the database based on their semantic similarity to the image, and then assigns to the image the best matching candidate category according to the same vision-language model. Experiments on benchmark datasets validate that CaSED outperforms other complex vision-language frameworks, while being efficient with much fewer parameters, paving the way for future research in this direction[1].

## 1   Introduction

Large-scale Vision-Language Models (VLMs) [47, 62, 34] enabled astonishing progress in computer vision by aligning multimodal semantics in a shared embedding space. This paper focuses on their use for image classification, where models such as CLIP [47] demonstrated strength in zero-shot transfer. While we witnessed advances in VLM-based classification in many directions, *e.g.* prompt learning [53, 66], scaling up to larger models and datasets [25, 45, 6], or by jointly considering captioning task [62, 34], they all assume a finite set of target categories, *i.e.* the *vocabulary*, to be pre-defined and static (as shown in Fig. 1a). However, this assumption is fragile, as it is often violated in practical applications (*e.g.* robotics, autonomous driving) where semantic categories can either differ from the development/training to the deployment/testing or evolve dynamically over time.

In this work, we remove this assumption and study the new task of Vocabulary-free Image Classification (VIC). The objective of VIC is to assign an image to a class that belongs to an unconstrained language-induced semantic space at test time, *without a vocabulary*, *i.e.* without a pre-defined set of categories (as shown in Fig. 1b). The unconstrained nature of the semantic space makes VIC a challenging problem. First, the search space is extremely large, with a cardinality on the order

---

[1]Code and demo is available at `https://github.com/altndrr/vic`

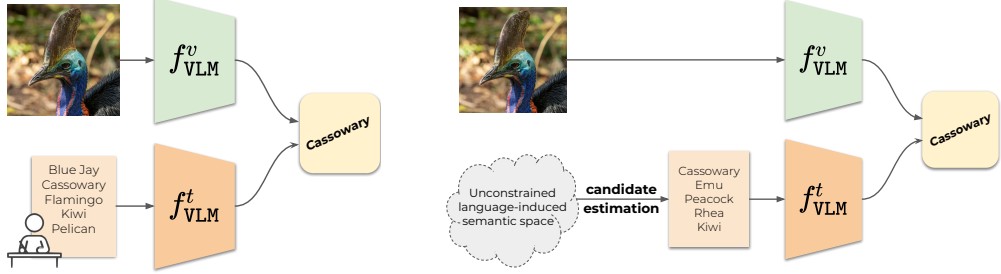

(a) VLM-based classification   (b) Vocabulary-free Image Classification

Figure 1: Vision-Language Model (VLM)-based classification (a) assumes a pre-defined set of target categories, *i.e.* the vocabulary, while our novel task (b) lifts this assumption by directly operating on the unconstrained language-induced semantic space, without a known vocabulary. $f_{\text{VLM}}^v$ and $f_{\text{VLM}}^t$ denote the pre-trained vision and text models of a VLM, respectively.

of millions of semantic concepts[2], way larger than any existing image classification benchmark (*e.g.* ImageNet-21k [10]). Second, it also includes very fine-grained concepts that might be hard to discriminate by the model. Third, as the categories encountered at test time are undefined beforehand, VIC calls for classification methods that do not rely on any vocabulary-aware supervision.

Recent web-scale Vision-Language Databases (VLDs) [51, 54], offer a unique opportunity to address VIC, as they cover a wide set of semantic concepts, ranging from general to highly specific ones. We empirically show that such external databases allow identifying semantic content that is more relevant to the target category than the captioning-enhanced VLMs supporting visual queries (*e.g.* BLIP-2 [33]). Motivated by this observation, we propose a training-free method for VIC, Category Search from External Databases (CaSED), which jointly leverages the discriminative multimodal representations derived from CLIP [47] and the information provided by recent VLDs (*e.g.* PMD [54]). CaSED operates in two steps: it first coarsely estimates a set of candidate categories for the test image, then it predicts the final category via multimodal matching. Specifically, we first retrieve the captions from the database that are semantically closer to the input image, from which we extract candidate categories via text parsing and filtering. We then estimate the similarity score between the input image and each candidate category via CLIP, using both visual and textual information, predicting as output the best matching candidate. CaSED exploits the pre-trained CLIP without further training, thus being flexible and computationally efficient.

We experiment on several datasets, considering both coarse- (*e.g.* Caltech-101 [14], UCF101 [55]) and fine-grained (*e.g.* FGVC-Aircraft [40], Flowers-102 [43]) classification tasks. To quantitatively assess the performance of methods addressing VIC, we also propose a set of metrics to measure how well the predicted class matches the semantics of the ground-truth label. Across all tasks and metrics, CaSED consistently outperforms all baselines, including VLMs as complex as BLIP-2 [33]. We believe that thanks to its simplicity and effectiveness, our training-free CaSED can serve as a competitive baseline for future works aiming to address the challenging VIC task.

To summarize, this work provides the following contributions:

- We explore the task of Vocabulary-free Image Classification  where the goal is to assign a class to an image over an unconstrained set of semantic concepts, overcoming the fundamental assumption of existing VLM-based methods for image classification. We formalize this task and suggest specific evaluation metrics that can be used as a reference for future works.
- We propose CaSED, the first method to address VIC thanks to the adoption of large captioning databases. Notably, CaSED is training-free, not requiring any additional parameter nor finetuning of the network's textual and visual encoders.
- Our large-scale evaluation demonstrates that CaSED consistently outperforms a more complex VLM such as BLIP-2 [33] on VIC, while requiring much fewer parameters.

---

[2]We estimate the number of concepts using BabelNet [42], which is close to 4 million for English.

## 2 Related work

**Vision-Language Models.** Leveraging large-scale datasets with image-text pairs [51, 50, 54], recent works train models by mapping the two modalities into a shared representation space [28, 18, 11, 47, 26, 35, 15]. A notable example is CLIP [47] which, using modality-specific encoders and a contrastive objective to align their output representations, showed remarkable performance on zero-shot classification. Subsequent works improved CLIP by *e.g.* connecting the two modalities via cross-modal attention [35], multi-object representation alignment [64], learning from weak-supervision [60] or unaligned data [54].

Another line of works improved vision-language pre-training for complex vision-language tasks, such as image captioning and visual question answering (VQA) [62, 22, 34, 1]. In this context, BLIP [34] exploits web data and generated captions to supervise pre-training of a multimodal architecture, outperforming existing VLMs on both captioning and VQA. The current state-of-the-art method BLIP-2 [33] trains a module connecting the two modalities on top of a frozen visual encoder and a frozen large-language model, enabling instructed zero-shot image-to-text generation.

In this work, we challenge a fundamental assumption of zero-shot classification with VLMs: the set of target classes is known a priori. We propose a new task, VIC, which sidesteps this assumption, performing classification in a language-induced open-ended space of semantic categories. We show that even BLIP-2 struggles in this scenario while external multimodal databases provide valuable priors for inferring the semantic category of an image. As a final note, VIC differs from open-vocabulary recognition (e.g. [63, 17]) since the latter assumes that the list of target classes is known and available to the model during inference.

**Retrieval augmented models.** In natural language processing, multiple works showed the benefit of retrieving information from external databases, improving the performance of large language models [19, 32, 3]. In computer vision, such a paradigm has been used mostly to deal with the imbalanced distribution of classes. Examples are [38, 39], addressing long-tail recognition by learning to retrieve training samples [38] or image-text pairs from an external database [39]. Similarly, [58] retrieves images from a given dataset to learn fine-grained visual representations. More recently, retrieval-augmentation has been extended to various types of sources for visual question answering [23], as well as to condition the generative process in diffusion models [2], or image captioning [48]. Our work is close in spirit to [39], as we exploit an external database. However, [39] assumes a pre-defined set of classes (and data) available for training a retrieval module, something we cannot have for the extremely large semantic space of VIC. In CaSED, retrieval is leveraged to first create a set of candidate classes, and to then perform the final class prediction. Moreover, we assume the database to contain only captions, and not necessarily paired image-text data, thus being less memory-demanding.

Finally, the performance of retrieval models is largely affected by the employed database. In the context of VLMs, researchers collected multiple open datasets to study vision-language pre-training. Two notable examples are LAION-5B [51], collected by filtering Common Crawl [8] via CLIP, and the Public Multimodal Datasets (PMD) [54], collecting image-text pairs from different public datasets, such as Conceptual Captions [52, 5], YFCC100M [57], Wikipedia Image Text [56], and Redcaps [12]. In our experiments, we use a subset of PMD as database and investigate how classification performance varies based on the size and quality of the database.

## 3 Vocabulary-free Image Classification

**Preliminaries.** Given the image space $\mathcal{X}$ and a set of class labels $C$, a classification model $f : \mathcal{X} \rightarrow C$ is a function mapping an image $\boldsymbol{x} \in \mathcal{X}$ to its corresponding class $\boldsymbol{c} \in C$. While $C$ may be represented by indices that refer to semantic classes, here we focus on cases where $C$ consists of a set of concept names that correspond to real entities. Note that $C$ is a finite set whose elements belong to the semantic space $\mathcal{S}$, *i.e.* $C \subset \mathcal{S}$. In standard image classification, $C$ is given a priori, and $f$ is learned on a dataset of image-label pairs.

Recently, the introduction of contrastive-based VLMs [47, 25], which learn a cross-modal aligned feature space, revised this paradigm by defining a function $f_{\texttt{VLM}}$ that infers the similarity between an

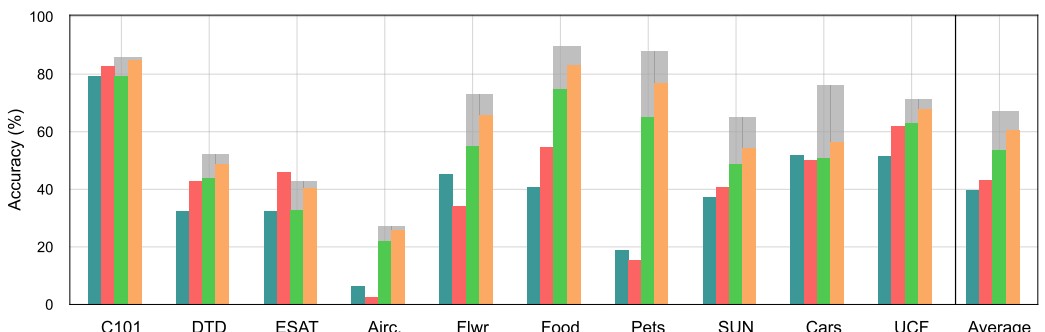

Figure 2: Results of our preliminary study, showing the top-1 accuracy when matching semantic descriptions to ground-truth class names in ten different datasets. We compare BLIP-2 (VQA) and BLIP-2 (Captioning) with Closest Caption and Captions Centroid, *i.e.* the average representation of the retrieved captions. We additionally highlight the Upper bound for zero-shot CLIP. Representing the large semantic space as VLDs and retrieving captions from it produces semantically more similar outputs to ground-truth labels w.r.t. querying outputs from VQA-enabled VLMs, while requiring 10 times fewer parameters compared to the latter.

image and a textual description $t \in \mathcal{T}$, *i.e.* $f_{\text{VLM}} : \mathcal{X} \times \mathcal{T} \rightarrow \mathbb{R}$, with $\mathcal{T}$ being the language space. Given this function, classification can be performed with:

$$f(\boldsymbol{x}) = \arg \max_{\boldsymbol{c} \in C} f_{\text{VLM}}(\boldsymbol{x}, \phi(\boldsymbol{c})) \tag{1}$$

where $\phi(\boldsymbol{c})$ is a string concatenation operation, combining a fixed text template, *i.e.* a *prompt*, with a class name. This definition allows for zero-shot transfer, *i.e.* performing arbitrary classification tasks by re-defining the set $C$ at test time, without re-training the model. However, such zero-shot transfer setup still assumes that the set $C$ is provided. In this work, Vocabulary-free Image Classification (VIC) is formulated to surpass this assumption.

**Task definition.** VIC aims to assign a class $\boldsymbol{c}$ to an image $\boldsymbol{x}$ *without* prior knowledge on $C$, thus operating on the semantic class space $\mathcal{S}$ that contains all the possible concepts. Formally, we want to produce a function $f$ mapping an image to a semantic label in $\mathcal{S}$, *i.e.* $f : \mathcal{X} \rightarrow \mathcal{S}$. Our task definition implies that at test time, the function $f$ has only access to an input image $\boldsymbol{x}$ and a large source of semantic concepts that approximates $\mathcal{S}$. VIC is a challenging classification task by definition due to the extremely large cardinality of the semantic classes in $\mathcal{S}$. As an example, ImageNet-21k [10], one of the largest classification benchmarks, is 200 times smaller than the semantic classes in BabelNet [42]. This large search space poses a prime challenge for distinguishing fine-grained concepts across multiple domains as well as ones that naturally follow a long-tailed distribution.

**Semantic space representation.** As the main challenge of VIC, how to represent the large semantic space plays a fundamental role in the method design. We can either model the multimodal semantic space directly with a VLM equipped with an autoregressive language decoder [34] or via image-text retrieval from VLDs. Consequently, we can approach VIC either via VQA-enabled VLMs by querying for the candidate class given the input image, or by retrieving and processing data from an external VLD to obtain the candidate class.

To investigate the two potential strategies, we perform a preliminary experimental analysis to understand how well the output of a method semantically captures the image category, or in other words, to assess the alignment of class boundaries in the visual and textual representations. Specifically, we compare the semantic accuracy of querying VQA VLMs and of retrieving from VLDs w.r.t. the ground-truth class labels. We consider the output of a method as correct if its closest textual embedding among the target classes of the dataset corresponds to the ground-truth class of the test sample[3]. We exploit the text encoder of CLIP (ViT-L) [47] to obtain textual embeddings.

---

[3]Note that this metric is not the standard accuracy in image classification as we use distances in the embedding space to ground predictions from the unconstrained semantic space to the set of classes in a specific dataset.

Regarding experimented methods, we select BLIP-2 [33] to represent VQA-enabled VLMs for its state-of-the-art performance in VQA benchmarks, while we use a subset of PMD [54] as the VLD. In particular, we compare the following methods: i) *BLIP-2 VQA*, which directly queries BLIP-2 for the image category; ii) *BLIP-2 Captioning*, which queries BLIP-2 for the image caption; iii) *Closest Caption*, which is the closest caption to the image, as retrieved from the database; iv) *Caption Centroid*, which averages the textual embeddings of the 10 most similar captions to the input image. As we use CLIP embeddings, if visual and textual representations perfectly align, the performance would be the same as zero-shot CLIP with given target classes. We thus report zero-shot CLIP to serve as the upper bound for retrieval accuracy.

We experiment on a variety of test datasets for both coarse- and fine-grained classification (see details in Sec. 5), and report the results in Fig. 2. The average textual embedding of the retrieved captions (*i.e.* Caption Centroid) achieves the best semantic accuracy for 9 datasets out of 10, consistently surpassing methods based on BLIP-2. On average, the accuracy achieved by Caption Centroid is $60.47\%$, which is $+17.36\%$ higher than the one achieved by BLIP-2 Captioning ($43.11\%$). Moreover, Captions Centroid achieves results much closer to the CLIP upper bound ($67.17\%$) than the other approaches. Notably, such VLD-based retrieval is also computationally more efficient, faster (~ 4 second for a batch size of 64 on a single A6000 GPU), and requires fewer parameters (approximately 10 times less) than BLIP-2 (see Tab. 7 in Appendix B).

The results of this preliminary study clearly suggest that representing the large semantic space with VLDs can produce (via retrieval) semantically more relevant content to the input image, in comparison to querying VQA-enabled VLMs, while being computationally efficient. Based on this conclusion, we develop an approach, Category Search from External Databases (CaSED), that searches for the semantic class from the large semantic space represented in the captions of VLDs.

## 4 CaSED: Category Search from External Databases

Our proposed method CaSED finds the best matching category within the unconstrained semantic space by multimodal data from large VLDs. Fig. 3 provides an overview of our proposed method. We first retrieve the semantically most similar captions from a database, from which we extract a set of candidate categories by applying text parsing and filtering techniques. We further score the candidates using the multimodal aligned representation of the large pre-trained VLM, *i.e.* CLIP [47], to obtain the best-matching category. We describe in detail each process in the following.

### 4.1 Generating candidate categories

We first restrict the extremely large classification space to a few most probable candidate classes. Let $f_{\text{VLM}}$ be the pre-trained VLM and $D$ be the external database of image captions. Given an input $\boldsymbol{x}$, we retrieve the set $D_{\boldsymbol{x}} \subset D$ of $K$ closest captions to the input image via

$$D_{\boldsymbol{x}} = \underset{\boldsymbol{d} \in D}{\text{top-k}} \; f_{\text{VLM}}(\boldsymbol{x}, \boldsymbol{d}) = \underset{\boldsymbol{d} \in D}{\text{top-k}} \; \langle f_{\text{VLM}}^{v}(\boldsymbol{x}), f_{\text{VLM}}^{t}(\boldsymbol{d}) \rangle, \tag{2}$$

where $f_{\text{VLM}}^{v} : \mathcal{X} \to \mathcal{Z}$ is the visual encoder of the VLM, $f_{\text{VLM}}^{t} : \mathcal{T} \to \mathcal{Z}$ is the textual encoder, and $\mathcal{Z}$ is their shared embedding space. The operation $\langle \cdot, \cdot \rangle$ indicates the computation of the cosine similarity. Note that our approach is agnostic to the particular form of $D$, and that it can accommodate a flexible database size by including captions from additional resources.

From the set $D_{\boldsymbol{x}}$, we then extract a finite set of candidate classes $C_{\boldsymbol{x}}$ by performing simple text parsing and filtering techniques, *e.g.* stop-words removal, POS tagging. Details on the filtering procedure can be found in Appendix A.

### 4.2 Multimodal candidate scoring

Among the small set $C_{\boldsymbol{x}}$, we score each candidate by accounting for both visual and textual semantic similarities using the VLM encoders, in order to select the best-matching class for the input image.

**Image-to-text score.** As the image is the main driver for the semantic class we aim to recognize, we use the visual information to score the candidate categories. We denote $s_{\boldsymbol{c}}^{v}$ as the visual score of each

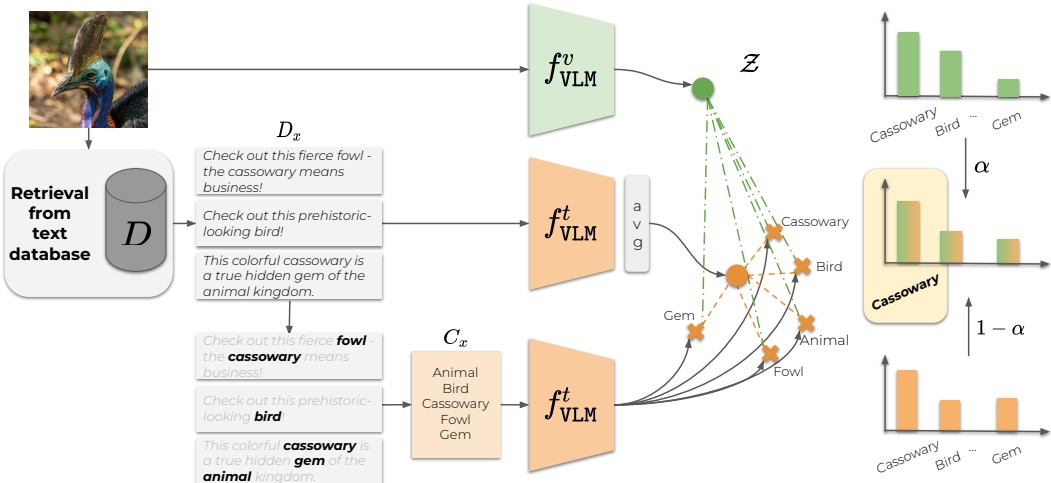

Figure 3: Overview of CaSED. Given an input image, CaSED retrieves the most relevant captions from an external database filtering them to extract candidate categories. We classify image -to- text and text -to- text , using the retrieved captions centroid as the textual counterpart of the input image.

candidate category $c$ and compute it as the similarity between the visual representation of the input image and the textual representation of the candidate name:

$$s_{c}^{v} = \langle f_{\text{VLM}}^{v}(\boldsymbol{x}), f_{\text{VLM}}^{t}(\boldsymbol{c}) \rangle. \tag{3}$$

The higher value of $s_{c}^{v}$ indicates a closer alignment between the target image and the candidate class.

**Text-to-text score.** While the image-to-text score $s_{c}^{v}$ is effective, there exists a well-known modality gap in the space $\mathcal{Z}$, harming the performance of zero-shot models [36]. As suggested by Fig. 2 in Sec. 3, the semantic relevance of the retrieved captions, and their centroid in particular, is high w.r.t. the underlying ground-truth label. We are therefore motivated to exploit such attributes and introduce a unimodal text-to-text scoring to mitigate the modality gap of cross-modal scoring.

Formally, we define the centroid $\bar{\boldsymbol{d}}_{\boldsymbol{x}}$ of the retrieved captions as:

$$\bar{\boldsymbol{d}}_{\boldsymbol{x}} = \frac{1}{K} \sum_{\boldsymbol{d} \in D_{\boldsymbol{x}}} f_{\text{VLM}}^{t}(\boldsymbol{d}), \tag{4}$$

where $K$ is the number of retrieved captions. We then define the text-based matching score $s_{c}^{t}$ as the similarity between the centroid and the candidate category:

$$s_{c}^{t} = \langle \bar{\boldsymbol{d}}_{\boldsymbol{x}}, f_{\text{VLM}}^{t}(\boldsymbol{c}) \rangle. \tag{5}$$

A higher value of $s_{c}^{t}$ means a higher alignment between the caption centroid and the candidate embedding. Note that the semantic relevance of the caption centroid is an inherent property of the text encoder of VLMs (*i.e.* CLIP). Since CLIP is trained with an image-text alignment loss, its text encoder focuses on the visual elements of the caption, discarding parts that are either non-visual or non-relevant to the visual content. This improves the model's robustness to noisy candidates.

**Final predicted candidate.** To predict the final candidate, we merge the two scores, obtaining the final score $s_{c}$ for each candidate $c$ as:

$$s_{c} = \alpha\,\sigma(s_{c}^{v})\ +\ (1 - \alpha)\,\sigma(s_{c}^{t}) \tag{6}$$

where $\sigma(\cdot)$ is the softmax operation on the two scores of each candidate class, and $\alpha$ is a hyperparameter regulating the contribution of the two modalities. Finally we obtain the output category as $f(x) = \arg\max_{\boldsymbol{c} \in C_{\boldsymbol{x}}} s_{\boldsymbol{c}}$. Notably, CaSED respects the VIC task definition, performing classification without known class priors, while being *training-free* with the use of a pre-trained and frozen VLM. This makes the approach flexible and applicable to a variety of architectures and databases.

# 5   Experiments

We evaluate CaSED in comparison to other VLM-based methods on the novel task Vocabulary-free Image Classification with extensive benchmark datasets covering both coarse-grained and fine-grained classification. We first describe the experimental protocol in terms of the datasets, the proposed evaluation metrics, and the baselines applicable to this task (Sec. 5.1). We then discuss the quantitative results regarding the comparison between our method and baselines (Sec. 5.2). Finally, we present a thorough ablation to justify the design choices of CaSED (Sec. 5.3). In addition, we provide a cost comparison between the baseline methods and CaSED in Appendix B. We also offer further ablation of our method in Appendix C regarding architecture and database, and showcase qualitative results of predicted categories from multiple datasets in Appendix D.

## 5.1   Experimental protocol

**Datasets.** We follow existing works [53, 66] and use ten datasets that feature both coarse-grained and fine-grained classification in different domains: Caltech-101 (C101) [14], DTD [7], EuroSAT (ESAT) [21], FGVC-Aircraft (Airc.) [40], Flowers-102 (Flwr) [43], Food-101 (Food) [4], Oxford Pets (Pets), Stanford Cars (Cars) [29], SUN397 (SUN) [61], and UCF101 (UCF) [55]. Additionally, we used ImageNet [10] for hyperparameters tuning.

**Evaluation metrics.** Due to the unconstrained nature of the semantic space, evaluating the effectiveness of methods for VIC is not trivial. In this paper, we propose to use two main criteria, namely *semantic relevance*, *i.e.* the similarity of the predicted class w.r.t. the ground-truth label, and *image grouping*, *i.e.* the quality of the predicted classes for organizing images into clusters. For semantic, we consider i) *Semantic Similarity*, *i.e.* the similarity of predicted/ground-truth labels in a semantic space, and ii) *Semantic IoU*, *i.e.* the overlap of words between the prediction and the true label. More formally, given an input $x$ with ground-truth label $y$ and prediction $\hat{c} = f(x)$, we compute the *Semantic Similarity* as $\langle g(\hat{c}), g(y) \rangle$, where $g : \mathcal{T} \to \mathcal{Y}$ is a function mapping text to an embedding space $\mathcal{Y}$. Since we want to model free-form text, we use Sentence-BERT [49] as $g$. For *Semantic IoU*, given a predicted label $c$, and assuming $c$ being a set of words, we compute the Semantic IoU as $|c \cap y|/|c \cup y|$, where $y$ is the set of words in the ground-truth label. To assess grouping, we measure the classic *Cluster Accuracy* by first clustering images according to their predicted label, and then assigning each cluster to a ground-truth label with Hungarian matching. Sometimes, this mapping is resolved with a many-to-one match, where a predicted cluster is assigned to the most present ground-truth label. This evaluation draws inspiration from the protocols used for deep visual clustering [59, 24, 20].

**Baselines.** We consider three main groups of baselines for our comparisons. The most straightforward baselines consist of using CLIP with large vocabularies, such as WordNet [41] (117k names) or the English Words (234k names [16]). As an upper bound, we also consider CLIP with the perfect vocabulary, *i.e.* the ground-truth names of the target dataset (CLIP upper bound). Due to lack of space, we only report results for CLIP with ViT-L [13], while results with other architectures are reported in Appendix C. The second group of baselines consists of captioning methods, as captions can well describe the semantic content of images. We consider two options: captions retrieved from a database and captions generated by a pre-trained image captioning model. For the former we exploit a large collection of textual descriptions, retrieving the most fitting caption for the given image. For the latter, we exploit BLIP-2 [33] — a VLM with remarkable performance on a variety of tasks, including image captioning — to provide a description for the image. The last group of baselines consists of using a VQA model to directly predict the class name associated with the image. Again, we consider BLIP-2 [33], since being highly effective also in VQA. We evaluate BLIP-2 over both ViT-L and ViT-g throughout the experiments [4].

**Implementation details.** Our experiments were conducted using NVIDIA A6000 GPUs with mixed-bit precision. As database, we use a subset of PMD [54], containing five of its largest datasets: Conceptual Captions (CC3M) [52], Conceptual Captions 12M (CC12M) [5], Wikipedia Image Text (WIT) [56], Redcaps [12], and a subset of [57] used for PMD (YFCC100M*). Further details on the

---

[4] Following the BLIP-2 [33] demo, for captioning, we used the prompt "Question: what's in the image? Answer:". For VQA, we used "Question: what's the name of the object in the image? Answer: a".

| Method | | Cluster Accuracy (%) ↑ | | | | | | | | | | |
|---|---|---|---|---|---|---|---|---|---|---|---|---|
| | | C101 | DTD | ESAT | Airc. | Flwr | Food | Pets | SUN | Cars | UCF | Avg. |
| CLIP | WordNet | 34.0 | 20.1 | 16.7 | 16.7 | 58.3 | 40.9 | 52.0 | 29.4 | 18.6 | 39.5 | 32.6 |
| | English Words | 29.1 | 19.6 | 22.1 | 15.9 | 64.0 | 30.9 | 44.4 | 24.2 | 19.3 | 34.5 | 30.4 |
| Caption | Closest Caption | 12.8 | 8.9 | 16.7 | 13.3 | 28.5 | 13.1 | 15.0 | 8.6 | 20.0 | 17.8 | 15.5 |
| | BLIP-2 (ViT-L) | 26.5 | 11.7 | 23.3 | 5.4 | 23.6 | 12.4 | 11.6 | 19.5 | 14.8 | 25.7 | 17.4 |
| | BLIP-2 (ViT-g) | 37.4 | 13.0 | **25.2** | 10.0 | 29.5 | 19.9 | 15.5 | 21.5 | 27.9 | 32.7 | 23.3 |
| VQA | BLIP-2 (ViT-L) | 60.4 | 20.4 | 21.4 | 8.1 | 36.7 | 21.3 | 14.0 | 32.6 | 28.8 | 44.3 | 28.8 |
| | BLIP-2 (ViT-g) | **62.2** | 23.8 | 22.0 | 15.9 | 57.8 | 33.4 | 23.4 | 36.4 | **57.2** | **55.4** | 38.7 |
| CaSED | | 51.5 | **29.1** | 23.8 | **22.8** | **68.7** | **58.8** | **60.4** | **37.4** | 31.3 | 47.7 | **43.1** |
| CLIP upper bound | | 87.6 | 52.9 | 47.4 | 31.8 | 78.0 | 89.9 | 88.0 | 65.3 | 76.5 | 72.5 | 69.0 |

Table 1: Cluster Accuracy on the ten datasets. Green is our method, gray shows the upper bound.

| Method | | Semantic Similarity (x100) ↑ | | | | | | | | | | |
|---|---|---|---|---|---|---|---|---|---|---|---|---|
| | | C101 | DTD | ESAT | Airc. | Flwr | Food | Pets | SUN | Cars | UCF | Avg. |
| CLIP | WordNet | 48.6 | 32.7 | 24.4 | 18.9 | 55.9 | 49.6 | 53.7 | 44.9 | 28.8 | 44.2 | 40.2 |
| | English Words | 39.3 | 31.6 | 19.1 | 18.6 | 43.4 | 38.0 | 44.2 | 36.0 | 19.9 | 34.7 | 32.5 |
| Caption | Closest Caption | 42.1 | 23.9 | 23.4 | 29.2 | 40.0 | 46.9 | 40.2 | 39.8 | 49.2 | 40.3 | 37.5 |
| | BLIP-2 (ViT-L) | 57.8 | 31.4 | **39.9** | 24.4 | 36.1 | 44.6 | 29.0 | 45.3 | 46.4 | 38.0 | 39.3 |
| | BLIP-2 (ViT-g) | 63.0 | 33.1 | 36.2 | 24.3 | 45.2 | 51.6 | 31.6 | 48.3 | 61.0 | 44.6 | 43.9 |
| VQA | BLIP-2 (ViT-L) | 70.5 | 34.9 | 29.7 | 29.1 | 48.8 | 42.0 | 40.0 | 50.6 | 52.4 | 48.6 | 44.7 |
| | BLIP-2 (ViT-g) | **73.5** | 36.5 | 31.4 | **30.8** | 59.9 | 52.1 | 43.9 | **53.3** | **65.1** | 55.1 | 50.1 |
| CaSED | | 65.7 | **40.0** | 32.0 | 30.3 | 55.5 | **64.5** | **62.5** | 52.5 | 47.4 | 54.1 | **50.4** |
| CLIP upper bound | | 90.8 | 69.8 | 67.7 | 66.7 | 83.4 | 93.7 | 91.8 | 80.5 | 92.3 | 83.3 | 82.0 |

Table 2: Semantic Similarity on the ten datasets. Values are multiplied by x100 for readability. Green highlights our method and gray indicates the upper bound.

selection are left for Appendix C. We speed up the retrieval process by embedding the database via the text encoder $f_{\text{VLM}}^t$ and using fast indexing technique, *i.e.* FAISS [27]. We tuned the $\alpha$ hyperparameter of Eq. (6) and the number of retrieved captions $K$ of our method on the ImageNet dataset, finding that $\alpha = 0.7$ and $K = 10$ led to the best results. We use these values across all experiments.

## 5.2 Quantitative results

The results of CaSED and the baselines are presented in Table 1, Table 2, and Table 3 for Cluster Accuracy (%), Semantic Similarity, and Semantic IoU (%), respectively. CaSED consistently outperforms all baselines in all metrics on average and in most of the datasets. Notably, CaSED surpasses BLIP-2 (VQA) over ViT-g by $+4.4\%$ in Cluster Accuracy and $+1.7\%$ on Semantic IoU, while using much fewer parameters (*i.e.* 102M vs 4.1B). The gap is even larger over the same visual backbone, with CaSED outperforming BLIP-2 on ViT-L (VQA) by $+14.3\%$ on Cluster Accuracy, $+5.7$ in Semantic Similarity, and $+6.8\%$ on Semantic IoU. These results highlight the effectiveness of CaSED, achieving the best performance both in terms of semantic relevance and image grouping.

An interesting observation from the tables is that simply applying CLIP on top of a pre-defined, large vocabulary, is not effective in VIC. This is due to the challenge of classifying over a huge search space, where class boundaries are hard to model. This is confirmed by the results of CLIP with English Words having a larger search space but performing consistently worse than CLIP with WordNet across all metrics (*e.g.* $-7.7$ on Semantic Similarity, -3.2% on Semantic IoU).

A final observation relates to the captioning models. Despite their ability to capture the image semantic even in challenging settings (*e.g.* 39.3 Semantic Similarity of BLIP-2 ViT-L on ESAT), captions exhibit high variability across images of the same category. This causes the worst performance on Clustering and Semantic IoU across all approaches (*e.g.* almost $-20\%$ less than CaSED on average in terms of Cluster Accuracy), thus demonstrating that while captions can effectively describe the content of an image, they do not necessarily imply better categorization for VIC.

| Method | | C101 | DTD | ESAT | Airc. | Semantic IoU (%) ↑ Flwr | Food | Pets | SUN | Cars | UCF | Avg. |
|---|---|---|---|---|---|---|---|---|---|---|---|---|
| CLIP | WordNet | 15.0 | 3.0 | 1.3 | 0.5 | 31.3 | 7.8 | 14.7 | 9.0 | 4.8 | 3.8 | 9.1 |
| | English Words | 8.0 | 2.0 | 0.0 | 1.1 | 16.4 | 2.0 | 17.2 | 8.1 | 2.7 | 1.8 | 5.9 |
| Caption | Closest Caption | 4.5 | 0.8 | 1.3 | 1.9 | 5.9 | 3.1 | 3.0 | 2.3 | 11.4 | 1.0 | 3.5 |
| | BLIP-2 (ViT-L) | 13.4 | 1.4 | 4.8 | 0.0 | 7.5 | 4.7 | 1.7 | 4.7 | 11.6 | 1.1 | 5.1 |
| | BLIP-2 (ViT-g) | 16.8 | 1.8 | 4.1 | 0.1 | 13.9 | 7.9 | 2.9 | 5.7 | 24.7 | 1.9 | 8.0 |
| VQA | BLIP-2 (ViT-L) | 36.1 | 1.8 | 7.0 | 0.1 | 21.5 | 3.7 | 5.7 | 11.5 | 18.9 | 2.5 | 10.9 |
| | BLIP-2 (ViT-g) | **41.5** | 2.4 | **7.5** | 2.0 | **38.0** | 8.6 | 10.2 | 13.8 | **33.2** | 2.8 | 16.0 |
| CaSED | | 35.4 | **5.1** | 2.3 | **4.8** | 33.1 | **19.4** | **35.1** | **17.2** | 16.2 | **8.4** | **17.7** |
| CLIP upper bound | | 86.0 | 52.2 | 51.5 | 28.6 | 75.7 | 89.9 | 88.0 | 66.6 | 84.5 | 71.3 | 69.4 |

Table 3: Semantic IoU on the ten datasets. Green is our method, gray shows the upper bound.

| Candidates Generation | Scoring Vis. | Lang. | CA | S-Sim. | S-IoU |
|---|---|---|---|---|---|
| Generative [33] | ✓ | | 23.3 | 47.1 | 11.9 |
| Retrieval | ✓ | | 41.7 | 49.3 | 17.0 |
| | | ✓ | 42.7 | 50.3 | 17.0 |
| | ✓ | ✓ | **43.1** | **50.4** | **17.7** |

(a) Ablation on candidate generation and scoring.

| Database | Size | CA | S-Sim. | S-IoU |
|---|---|---|---|---|
| CC3M | 2.8M | 34.2 | 47.9 | 13.1 |
| WIT | 4.8M | 34.6 | 42.9 | 12.1 |
| Redcaps | 7.9M | 42.0 | 49.5 | 17.2 |
| CC12M | 10.3M | **44.0** | **51.3** | **18.3** |
| YFCC100M* | 29.9M | 40.7 | 48.8 | 17.1 |
| All | 54.8M | 43.1 | 50.4 | 17.7 |

(b) Ablation on the database.

Table 4: Ablation studies. Metrics are averaged across the ten datasets. Green highlights our setting. **Bold** represents best, underline indicates second best.

## 5.3 Ablation studies

In this section, we present the ablation study associated with different components of our approach. We first analyze the impact of our retrieval-based candidate selection and multimodal scoring. We then show the results of CaSED for different databases, and how the number of retrieved captions impacts the performance.

**Candidates generation.** We consider two options to generate the set of candidate classes. The first uses BLIP-2 (ViT-g), asking for multiple candidate labels for the input image. The second is our caption retrieval and filtering strategy. Table 4a shows the results where, for fairness, we score both sets with the same VLM (*i.e.* CLIP ViT-L). Our approach consistently outperforms BLIP-2 across all metrics (*e.g.* 41.7 vs 23.3 for Cluster Accuracy). This confirms the preliminary results of our study in Fig. 2, with retrieved captions providing better semantic priors than directly using a powerful VLM.

**Multimodal scoring.** The second ablation studies the effect of our proposed multimodal scoring vs its unimodal counterparts. As Table 4a shows, multimodal candidate scoring provides the best results across all metrics, with clear improvements over the visual modality alone (*i.e.* +1.4% of Cluster Accuracy). Notably, scoring via language only partially fills the gap between visual and multimodal scoring (*e.g.* +1% on Cluster Accuracy and +1 on Semantic Similarity), confirming that caption centroids contain discriminative semantic information.

**Retrieval database.** We analyze the impact of retrieving captions from different databases, using five public ones, *i.e.* CC3M, WIT, Redcaps, CC12M, and a subset of YFCC100M. The databases have different sizes (*e.g.* from 2.8M captions of CC3M to 29.9M captions of the YFCC100M subset), and different levels of noise. As shown in Table 4b, the results tend to improve as the size of the database increases (*e.g.* +8.9% on Cluster Accuracy over CC3M). However, the quality of the captions influences the performance, with CC12M and Redcaps alone achieving either comparable or slightly better results than the full database. These results suggest that while performance improves with the size of the database, the quality of the captions has a higher impact than the mere quantity.

**Number of captions.** Finally, we check how performance varies w.r.t. the number of retrieved captions. As shown in Fig. 4, all metrics consistently improve as the number of captions increases from 1 to 10. After that, performance tends to saturate, with a slight decrease in terms of Semantic Similarity for $K = 20$. These results suggest that while a minimum number of captions is needed to fully capture the semantics of the image, the possible interference of less-related (or even noisy) captions may impact the final performance. Future research may further improve the performance on VIC by focusing on how to deal with noisy retrieval results.

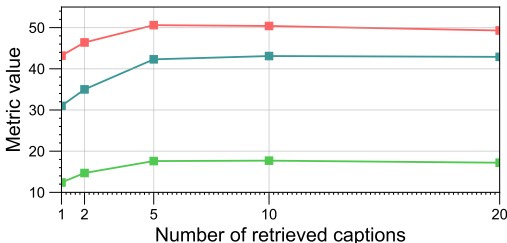

Figure 4: Ablation on the number of retrieved captions. We report Cluster accuracy (%), Semantic similarity, and Semantic IoU (%).

## 6  Discussion and conclusions

In this work, we proposed a new task, VIC, which operates on an unconstrained semantic space, without assuming a pre-defined set of classes, a brittle assumption of VLM-based classification. We experimentally verified that multimodal databases provide good semantic priors to restrict the large search space, and developed CaSED, an efficient training-free approach that retrieves the closest captions to the input image to extract candidate categories and scores them in a multimodal fashion. On different benchmarks, CaSED consistently achieved better results than more complex VLMs.

**Limitations and future works.** The performance of CaSED strongly depends on the choice of the retrieval database, with potential issues in retrieving concepts that are not well represented in the latter. Moreover, if the domain of application contains fine-grained concepts, a generic database might not be suitable. Without any prior information on the test labels, it is hard to predict the performance of a database a priori. On the other hand, CaSED can flexibly mitigate this issue by incrementally including new concepts in the database (even domain-specific ones) from textual corpora, without retraining. Future research may explore strategies to automatically select/extend a database based on test samples and/or pre-computing the best database to use in the absence of test label information.

Additionally, as CaSED lacks control over the classes contained in the database, its predictions might reflect potential biases. Improvements in mitigating biases and data quality control would reduce this issue. Another limitation is that CaSED does not keep track of its output history. This may lead to inconsistent predictions, *i.e.* assigning slightly different labels to images of the same semantic concept (*e.g. cassowary* vs *Casuarius*). Equipping CaSED with a memory storing the predicted labels may address this issue. Finally, CaSED does not deal with different class granularities: *e.g.* an image of a *cassowary* can be as well predicted as a *bird*. Future works may disambiguate such cases by explicitly integrating the user needs within VIC models.

**Broader impact.** CaSED addresses VIC in a scalable and training-free manner. We believe that our new problem formulation, metrics, and the effectiveness of CaSED will encourage future research in this topic, overcoming the limiting assumption of VLM-based classification and allowing the power of VLMs to benefit dynamic and unconstrained scenarios.

## Acknowledgements

This work was supported by the MUR PNRR project FAIR - Future AI Research (PE00000013) and ICSC National Research Centre for High Performance Computing, Big Data and Quantum Computing (CN00000013), funded by the NextGenerationEU. E.R. is partially supported by the PRECRISIS, funded by the EU Internal Security Fund (ISFP-2022-TFI-AG-PROTECT-02-101100539), the EU project SPRING (No. 871245), and by the PRIN project LEGO-AI (Prot. 2020TA3K9N). The work was carried out in the Vision and Learning joint laboratory of FBK and UNITN. We also thank the Deep Learning Lab of the ProM Facility for the GPU time.

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

# Appendix

## A   Candidates filtering

With the closest captions retrieved from the external database given the input image (please refer to Sec. 4 of the main manuscript for further details), we post-process them to filter out a set of candidate category names. We first create a set of all words that are contained in the captions. Then we apply sequentially three different groups of operations to the set to (i) remove noisy candidates, (ii) standardize their format, and (iii) filter them.

With the first group of operations, we remove all the irrelevant textual contents, such as tokens (*i.e.* "<PERSON>"), URLs, or file extensions. Note that, for the file extensions, we remove the extension but retain the file name as it might contain candidate class names. We also remove all the words that are shorter than three characters and split compound words by underscores or dashes. Finally, we remove all those terms containing symbols or numbers and meta words that are irrelevant to the classification task, such as "image", "photo", or "thumbnail". As shown in Table 5, when compared to having no operation for candidate filtering (first row), this set of operations removes inappropriate content and increases the accuracy of clusters by $+6.4\%$ and improves the semantic IoU by $+0.3$. However, we can observe a drop in semantic similarity by $-1.5$. This might be due to the removal of unnatural words that could still describe well the content of the image, *i.e.* underline- or dash-separated words, or URLs since they are longer w.r.t. natural words.

The second group of operations standardize the candidate names by aligning words that refer to the same semantic class to a standard format, reducing class redundancy. For example, "cassowary" and "Cassowary" will be considered as a single class instead of two. To this end, we perform two operations—lowercase conversion and singular form conversion. With such standardizing conversions, we observe a sizeable boost in terms of performances when compared to the results obtained by applying only removal-related operations. As shown in Table 5, we achieve higher results across all three metrics, leading to a relative improvement of $+7.2\%$, $+0.7$, and $+1.2$ in terms of cluster accuracy, semantic similarity, and semantic IoU, respectively.

The last group of operations considers two forms of filtering, where the first aims to filter out entire categories of words via Part-Of-Speech (POS) tag and the second aims to filter out rare and noisy contents based on the word occurrences. We select these two operations since common dataset class names do not contain terms that carry no semantics *e.g.* articles and pronouns, and since [9] showed that CLIP performs better when exposed to a smaller amount of unique tokens. The POS tagging[5] categorizes words into groups, such as adjectives, articles, nouns, or verbs, enabling us to filter all the terms that are not semantically meaningful for a classification task. Regarding the occurrence filtering, we first count how often a word appears in the retrieved captions and then we remove words that appear only once to make the candidate list less noisy. We can see from Table 5 that the inclusion of this final set of operations scores the best among all three metrics when compared to the results obtained when only the previous two groups of operations are applied.

**Number of captions vs number of selected candidates.**

To complement the previous analysis, in Table 6, we report the number of unique candidates extracted by the candidate filtering procedure, averaged over the ten datasets, and with an increasing number of retrieved captions, i.e., 1, 2, 5, 10, and 20. In the table, we show both the number of candidates extracted and the number of selected words. As the number of retrieved captions increases, the unique number of candidate words also increases, i.e., from 3849 with 1 caption to 28781 with 20. However, the number of selected words stabilizes around 800 as soon as we retrieve more than 1 caption. Having more captions reduces the noises in the selected words, something that might be present when relying on a single caption.

## B   Computational cost

We analyze the computational efficiency of CaSED versus BLIP-2 performing VQA and captioning, and report their respective number of parameters and inference time in Table 7. Notably, the methods

---

[5]We use the NLP library flair (`https://github.com/flairNLP/flair`).

| Operations | | | CA | S-Sim. | S-IoU |
|---|---|---|---|---|---|
| Remove | Standardize | Filter | | | |
| | | | 27.7 | 48.8 | 15.0 |
| ✓ | | | 34.1 | 47.3 | 15.3 |
| ✓ | ✓ | | 41.3 | 48.0 | 16.5 |
| ✓ | ✓ | ✓ | **43.1** | **50.4** | **17.7** |

Table 5: Ablation on the candidate filtering operations. Metrics are averaged across the ten datasets.

| Num. captions | CA | S-Sim. | S-IoU | Candidates | Selected |
|---|---|---|---|---|---|
| 1 | 30.9 | 43.2 | 12.4 | 3849 | 1047 |
| 2 | 35.0 | 46.4 | 14.6 | 6867 | 854 |
| 5 | 42.3 | **50.6** | 17.5 | 14548 | 775 |
| 10 | **43.1** | 50.4 | **17.7** | 22475 | 794 |
| 20 | 42.9 | 49.3 | 17.1 | 28781 | 802 |

Table 6: Extended ablation on the number retrieved captions. Green represents our selected configuration. We expand the results of Fig. 4 in the main manuscript to show the number of unique words extracted from captions (*i.e.*, "candidates") and the number of words selected by CaSED for classification (i.e., "selected"). Results are averaged across the ten datasets.

using external databases are consistently faster than BLIP-2. For instance, CaSED achieves a speed-up of 1.5x with respect to both the largest captioning model and the largest VQA model, while also achieving better performance. Overall, the fastest method is Closest Caption, which exploits the external database to retrieve a single caption and does not consider any candidate extraction pipeline. Conversely, our method retrieves the ten most similar captions and post-processes them to extract class name, resulting in a increase in inference time of approximately 3 times. Compared with the CLIP upper bound, our method considers multiple additional steps, each adding extra inference time. First, our method retrieves candidates from the external database to then extract the class names. Second, we have to forward the class names through the text encoder for each sample, while CLIP can forward them once and cache the features for later reuse.

When using an external database, note that an increase in database size implies a minimal variation in retrieval time. This is demonstrated by the computational cost required by retrieving from the large LAION-400M [51] database. As the results show, inference time is comparable between CaSED and CaSED (LAION-400M) despite the latter being approximately 8 times larger.

## C  Additional ablation study

In this section, we show the performance of our model with different backbones, and other commonly used databases for retrieval.

**Backbone architecture.** To answer this natural question about whether the outcome of our model depends on the backbone architecture, we further extend our main results with a CLIP model with a ResNet50 architecture. We report this additional ablation in Table 8, Table 9, and Table 10 for the cluster accuracy, the semantic IoU, and the semantic similarity, respectively. We can see that the performance with the CLIP ResNet50 is lower across all the metrics compared to CLIP ViT-L/14. This is expected since ResNet50 is a a smaller architecture, thus with a reduced capacity for semantic representation learning as compared to ViT-L/14. Nevertheless, our method with ResNet50 is still competitive against BLIP-2 models while using 40x fewer parameters (note that our ViT-L implementation uses 10x fewer parameters).

**Retrieval database.** We analyze the impact of retrieving captions from databases of different scales, expanding our main results with the four unused databases of PMD, including COCO [37], SBU Captions [44], Localized Narratives [46], and Visual Genome [30]. Moreover, we evaluate our method on other three databases, namely Ade20K [65], OpenImages [31], and LAION-400M [51]. These databases further extend the range of database sizes, with Ade20k containing only 0.07M captions and LAION-400M having 413M. We report the results in Table 11, ordering the rows by the database size. Differently from the tables reported in the main results, the results in Table 11

| | Method | Num. Params. | Inference time (ms) ↓ |
|---|---|---|---|
| | Closest Caption | 0.43B | 1390 ± 10 |
| Caption | BLIP-2 (ViT-L) | 3.46B | 5710 ± 153 |
| | BLIP-2 (ViT-g) | 4.37B | 6870 ± 177 |
| VQA | BLIP-2 (ViT-L) | 3.46B | 5670 ± 135 |
| | BLIP-2 (ViT-g) | 4.37B | 6650 ± 117 |
| | CaSED | 0.43B | 4370 ± 13 |
| | CaSED (LAION-400M) | 0.43B | 4350 ± 16 |
| | CLIP upper bound | 0.43B | 645 ± 77 |

Table 7: Computational cost of different methods. Green is our method, gray shows the upper bound. Inference time is reported on batches of size 64, as the average over multiple runs.

| | Method | | Cluster Accuracy (%) ↑ | | | | | | | | | | |
|---|---|---|---|---|---|---|---|---|---|---|---|---|---|
| | | | C101 | DTD | ESAT | Airc. | Flwr | Food | Pets | SUN | Cars | UCF | Avg. |
| **CLIP RN50** | CLIP | WordNet | 30.3 | 18.3 | **22.5** | 13.2 | 47.8 | 31.4 | 45.2 | 26.0 | 14.2 | 31.2 | 28.0 |
| | | English Words | 24.8 | 17.5 | 18.5 | 13.4 | 49.5 | 23.1 | 36.6 | 22.2 | 15.5 | 27.1 | 24.8 |
| | Caption | Closest Caption | 9.7 | 7.1 | 13.3 | 8.4 | 21.2 | 6.2 | 8.7 | 6.5 | 12.8 | 14.9 | 10.9 |
| | | CaSED | **44.6** | 23.9 | 12.5 | **15.3** | 58.8 | 48.7 | 50.1 | 32.8 | 24.6 | 33.9 | **34.5** |
| | | CLIP upper bound | 82.1 | 41.5 | 33.5 | 19.6 | 63.1 | 74.6 | 78.9 | 55.6 | 54.9 | 58.4 | 56.2 |
| **CLIP ViT-L/14** | CLIP | WordNet | 34.0 | 20.1 | 16.7 | 16.7 | 58.3 | 40.9 | 52.0 | 29.4 | 18.6 | 39.5 | 32.6 |
| | | English Words | 29.1 | 19.6 | 22.1 | 15.9 | 64.0 | 30.9 | 44.4 | 24.2 | 19.3 | 34.5 | 30.4 |
| | Caption | Closest Caption | 12.8 | 8.9 | 16.7 | 13.3 | 28.5 | 13.1 | 15.0 | 8.6 | 20.0 | 17.8 | 15.5 |
| | | CaSED | **51.5** | 29.1 | **23.8** | 22.8 | 68.7 | 58.8 | 60.4 | 37.4 | 31.3 | 47.7 | 43.1 |
| | | CLIP upper bound | 87.6 | 52.9 | 47.4 | 31.8 | 78.0 | 89.9 | 88.0 | 65.3 | 76.5 | 72.5 | 69.0 |
| | Caption | BLIP-2 (ViT-L) | 26.5 | 11.7 | 23.3 | 5.4 | 23.6 | 12.4 | 11.6 | 19.5 | 14.8 | 25.7 | 17.4 |
| | | BLIP-2 (ViT-g) | 37.4 | 13.0 | 25.2 | 10.0 | 29.5 | 19.9 | 15.5 | 21.5 | 27.9 | 32.7 | 23.3 |
| | VQA | BLIP-2 (ViT-L) | 60.4 | 20.4 | 21.4 | 8.1 | 36.7 | 21.3 | 14.0 | 32.6 | 28.8 | 44.3 | 28.8 |
| | | BLIP-2 (ViT-g) | 62.2 | 23.8 | 22.0 | 15.9 | 57.8 | 33.4 | 23.4 | 36.4 | 57.2 | 55.4 | 38.7 |

Table 8: Cluster Accuracy on the ten datasets. Green is our method, gray shows the upper bound. **Bold** represents best, underline indicates best considering also image captioning and VQA models.

are obtained on ImageNet. We first discuss the databases belonging to the PMD superset, and then analyze the results obtained with Ade20K, OpenImages, and LAION-400M. In the table, we also show the POS tag distribution (e.g. nouns, adjectives, verbs) of each dataset. Moreover, we report a metric, the semantic similarity to the closest caption in the dataset of each image (C.C. S-Sim) to check how close a database is to the target one (i.e. ImageNet).

By comparing the nine databases of PMD, we can perceive a non-uniform variation in performance across databases, with a remarkable gap between the top-5 (highlighted in blue) and the rest. The performance of the fifth-performing (i.e. WIT) and the sixth database (i.e. SBU Captions) differs by 12.8% on cluster accuracy, while the difference between the best-performing (i.e. CC12M) and the fifth is 10.9%. This observation motivates us to use the top-5 databases from PMD instead of the full superset to be efficient. We expand the best database with the other top-5 to ensure better coverage of class names in the textual descriptions.

With the additional databases, i.e. Ade20K, OpenImages, and LAION-400M, we further notice that there is a correlation between the size of a database and its performance. This was also noted in our results reported in the main manuscript. Ade20K, the smallest dataset, comprises only about 0.07M captions and obtains the lowest performance among all metrics. As the size of databases increase, the performance generally improves. However, it is important to note that dataset size alone is not the only requirement for good results. LAION-400M is of a much larger size than our split of PMD, yet the performance with LAION-400M is worse, which we attributes to the impact of much noisier retrieval from database of such scale. Instead, datasets such as CC12M and Redcaps perform better than LAION-400M despite being approximately 40x smaller. This finding suggests that the quality of the databases could be of a higher importance than its size.

| Method | | | Semantic IoU (%) ↑ | | | | | | | | | | |
|---|---|---|---|---|---|---|---|---|---|---|---|---|---|
| | | | C101 | DTD | ESAT | Airc. | Flwr | Food | Pets | SUN | Cars | UCF | Avg. |
| CLIP RN50 | CLIP | WordNet | 9.8 | 1.9 | **0.9** | 0.0 | 21.2 | 5.5 | 14.1 | 7.3 | 3.5 | 2.6 | 6.7 |
| | | English Words | 5.1 | 0.8 | 0.0 | 0.1 | 12.4 | 1.8 | 13.2 | 7.8 | 2.5 | 1.3 | 4.5 |
| | Caption | Closest Caption | 3.4 | 0.5 | 0.1 | 1.5 | 6.6 | 2.2 | 2.2 | 2.1 | 9.1 | 0.7 | 2.8 |
| | | CaSED | **31.1** | **2.9** | 0.08 | **2.8** | **29.7** | **15.1** | **27.6** | **15.0** | 13.4 | **5.2** | **14.3** |
| | | CLIP upper bound | 81.5 | 41.4 | 33.9 | 16.5 | 60.2 | 74.6 | 78.9 | 56.9 | 66.5 | 57.1 | 56.8 |
| CLIP ViT-L/14 | CLIP | WordNet | 15.0 | 3.0 | 1.3 | 0.5 | 31.3 | 7.8 | 14.7 | 9.0 | 4.8 | 3.8 | 9.1 |
| | | English Words | 8.0 | 2.0 | 0.0 | 1.1 | 16.4 | 2.0 | 17.2 | 8.1 | 2.7 | 1.8 | 5.9 |
| | Caption | Closest Caption | 4.5 | 0.8 | 1.3 | 1.9 | 5.9 | 3.1 | 3.0 | 2.3 | 11.4 | 1.0 | 3.5 |
| | | CaSED | **35.4** | **5.1** | **2.3** | **4.8** | **33.1** | **19.4** | **35.1** | **17.2** | 16.2 | **8.4** | **17.7** |
| | | CLIP upper bound | 86.0 | 52.2 | 51.5 | 28.6 | 75.7 | 89.9 | 88.0 | 66.6 | 84.5 | 71.3 | 69.4 |
| | Caption | BLIP-2 (ViT-L) | 13.4 | 1.4 | 4.8 | 0.0 | 7.5 | 4.7 | 1.7 | 4.7 | 11.6 | 1.1 | 5.1 |
| | | BLIP-2 (ViT-g) | 16.8 | 1.8 | 4.1 | 0.1 | 13.9 | 7.9 | 2.9 | 5.7 | 24.7 | 1.9 | 8.0 |
| | VQA | BLIP-2 (ViT-L) | 36.1 | 1.8 | 7.0 | 0.1 | 21.5 | 3.7 | 5.7 | 11.5 | 18.9 | 2.5 | 10.9 |
| | | BLIP-2 (ViT-g) | 41.5 | 2.4 | 7.5 | 2.0 | 38.0 | 8.6 | 10.2 | 13.8 | 33.2 | 2.8 | 16.0 |

Table 9: Semantic IoU on the ten datasets. Green is our method, gray shows the upper bound. **Bold** represents best, underline indicates best considering also image captioning and VQA models.

| Method | | | Semantic Similarity (x100) ↑ | | | | | | | | | | |
|---|---|---|---|---|---|---|---|---|---|---|---|---|---|
| | | | C101 | DTD | ESAT | Airc. | Flwr | Food | Pets | SUN | Cars | UCF | Avg. |
| CLIP RN50 | CLIP | WordNet | 43.2 | 29.0 | 18.5 | 21.6 | 46.7 | 44.6 | 50.3 | 42.8 | 26.4 | 40.0 | 36.3 |
| | | English Words | 36.0 | 29.5 | 14.9 | 20.0 | 38.1 | 34.2 | 40.7 | 35.4 | 18.3 | 32.4 | 29.9 |
| | Caption | Closest Caption | 37.2 | 22.8 | 14.2 | 26.8 | 38.9 | 41.2 | 32.6 | 37.4 | **44.3** | 32.4 | 32.8 |
| | | CaSED | **62.3** | **36.4** | **22.6** | **28.7** | **52.8** | **59.0** | **57.0** | **50.2** | 42.9 | **46.2** | **45.8** |
| | | CLIP upper bound | 88.1 | 62.4 | 52.8 | 53.9 | 72.0 | 83.7 | 85.8 | 73.9 | 81.0 | 73.9 | 72.7 |
| CLIP ViT-L/14 | CLIP | WordNet | 48.6 | 32.7 | 24.4 | 18.9 | 55.9 | 49.6 | 53.7 | 44.9 | 28.8 | 44.2 | 40.2 |
| | | English Words | 39.3 | 31.6 | 19.1 | 18.6 | 43.4 | 38.0 | 44.2 | 36.0 | 19.9 | 34.7 | 32.5 |
| | Caption | Closest Caption | 42.1 | 23.9 | 23.4 | 29.2 | 40.0 | 46.9 | 40.2 | 39.8 | **49.2** | 40.3 | 37.5 |
| | | CaSED | **65.7** | **40.0** | **32.0** | **30.3** | 55.5 | **64.5** | **62.5** | **52.5** | 47.4 | **54.1** | **50.4** |
| | | CLIP upper bound | 90.8 | 69.8 | 67.7 | 66.7 | 83.4 | 93.7 | 91.8 | 80.5 | 92.3 | 83.3 | 82.0 |
| | Caption | BLIP-2 (ViT-L) | 57.8 | 31.4 | 39.9 | 24.4 | 36.1 | 44.6 | 29.0 | 45.3 | 46.4 | 38.0 | 39.3 |
| | | BLIP-2 (ViT-g) | 63.0 | 33.1 | 36.2 | 24.3 | 45.2 | 51.6 | 31.6 | 48.3 | 61.0 | 44.6 | 43.9 |
| | VQA | BLIP-2 (ViT-L) | 70.5 | 34.9 | 29.7 | 29.1 | 48.8 | 42.0 | 40.0 | 50.6 | 52.4 | 48.6 | 44.7 |
| | | BLIP-2 (ViT-g) | 73.5 | 36.5 | 31.4 | 30.8 | 59.9 | 52.1 | 43.9 | 53.3 | 65.1 | 55.1 | 50.1 |

Table 10: Semantic similarity on the ten datasets. Green is ours, gray shows the upper bound. **Bold** represents best, underline indicates best considering also image captioning and VQA models.

Another observation pertains to the differences between captioning and visual question answering (VQA) datasets (*e.g.* COCO, Localized Narratives, and Visual Genome) w.r.t. image-text datasets collected from the web (*e.g.* CC12M, Redcaps, and LAION-400M). In general, it appears that textual descriptions from captioning and VQA datasets perform worse than datasets scraped from the web. This difference in performance may be due to the type of description provided, as the former often describes specific regions of the image, while the latter provides a general description of the image as a whole. Since our prime objective is to understand the class name of the subject in the image (whether it be a location, a pet, or a car model), captions describing secondary objects can lead to sub-optimal results.

Interestingly, we found that the average performance on the 10 datasets of a chosen textual database generally correlates with the accuracy on ImageNet and with the closest caption semantic similarity (C.C. S-Sim in Table 11). Examples are the subset of PMD and CC12M, achieving the best average results on the 10 datasets according to all metrics (Tab. 4b of the main paper) while being the best on the ImageNet validation set w.r.t. both the closest caption semantic similarity and the ViC metrics. The ImageNet validation set can thus be used as a proxy to pick the textual database in case of a lack of priors on the test set.

| Database | Size | CA | S-Sim. | S-IoU | Nouns (%) | Adjs (%) | Verbs (%) | Concepts | C.C. S-Sim. |
|----------|------|-----|--------|-------|-----------|----------|-----------|----------|-------------|
| Ade20K | 0.07M | 11.0 | 31.8 | 1.7 | 82.7 | 10.6 | 6.7 | 3.1K | 19.0 |
| COCO | 0.9M | 13.8 | 35.6 | 2.6 | 86.1 | 9.5 | 4.4 | 14.6K | 22.6 |
| SBU Captions | 1.0M | 20.1 | 40.8 | 5.7 | 98.0 | 1.0 | 0.9 | 29.3K | 26.9 |
| OpenImages | 1.4M | 16.8 | 37.8 | 4.2 | 85.4 | 10.0 | 4.5 | 21.4K | 24.8 |
| Loc. Narr. | 1.9M | 15.7 | 37.1 | 4.1 | 86.9 | 9.8 | 3.2 | 30.9K | 24.9 |
| CC3M | 2.8M | 37.9 | 53.9 | 16.2 | 57.9 | 23.6 | 18.5 | 28.4K | 35.2 |
| WIT | 4.8M | 32.9 | 47.9 | 14.5 | 99.8 | 0.05 | 0.07 | 163.7K | 30.0 |
| Visual Genome | 5.4M | 14.6 | 35.2 | 3.8 | 75.1 | 15.6 | 9.2 | 21.1K | 26.7 |
| Redcaps | 7.9M | 41.1 | 54.7 | 19.6 | 98.6 | 0.7 | 0.6 | 50.5K | 37.5 |
| CC12M | 10.3M | **43.8** | **57.4** | **21.2** | 96.6 | 1.8 | 1.5 | 36.7K | **38.7** |
| YFCC100M* | 29.9M | 39.1 | 53.5 | 18.9 | 99.7 | 0.2 | 0.1 | 179.6K | 37.8 |
| Ours | 54.8M | 41.5 | 55.7 | 20.4 | 99.8 | 0.05 | 0.05 | 334.1K | **38.7** |
| LAION-400M | 413.8M | 37.7 | 52.7 | 18.7 | 99.4 | 0.5 | 0.1 | 1.68M | 37.3 |

Table 11: Ablation on the databases on ImageNet. Green is our database, Blue shows the top-5 databases of PMD, which we used to create Ours. We also evaluate the semantic similarity of the closest caption to each image in the dataset (*i.e.*, C.C. S-Sim.). **Bold** represents best, while underline second best. YFCC100M* indicates we use the subset of the dataset used in PMD.

For what concerns the POS tag distribution, it is hard to extract any pattern that justifies the performance of different databases. For instance, CC3M is the only database without an extreme imbalance for nouns (i.e., 57.9% compared to >96% for the other in the top-5) while still achieving good performance. Moreover, while the number of concepts depends on the size of the database, there is no clear correlation between this value and the achieved scores. As an example, Localized Narratives has a comparable amount of concepts w.r.t. CC3M but a gap of >20% in cluster accuracy, and of 10 points in semantic similarity and IoU. Finally, it is worth noting that most of the datasets have an average of 12.6 words per caption, while Visual Genome has an average of only 5.1 words per caption. This characteristic may explain why Visual Genome behaves differently in terms of the database scaling rule of increasing performance with the database size.

# D    Qualitative results

Last, we report some qualitative results of our method applied on three different datasets, namely Caltech-101 (Fig. 5), Food101 (Fig. 6), and SUN397 (Fig. 7), where the first is coarse, and the last two are fine-grained, focusing on food plates and places respectively. For each, we present a batch of five images, where the first three represent success cases and the last two show interesting failure cases. Each sample shows the image we input to our method with the top-5 candidate classes.

From the results, we can see that for many success cases, our method not only generates the correct class name and selects it as the best matching label, but it also provides valid alternatives for classification. For example, the third image in Fig 5 or the second image in Fig 6, where CaSED provides the names "dessert" for the cheesecake and the label "bird" for the ibis. This phenomenon also happens in failure cases, where *e.g.* the last sample in Fig 6 provides both the name "pizza" and the name "margherita" for the dish, despite selecting the wrong name from the set.

Another interesting observation is that our method provides names for different objects in the same scene. For instance, the third and fourth samples in Fig 6 contain labels for both "guacamole" and "tortillas" for the first, and for "mozzarella", "insalata", and "balsamic" for the second. A further detail on the latter case is the ability of CLIP to reason in multiple languages since "insalata" translates to "salad" from Italian to English.

Regarding failure cases, it is interesting to note that the candidate names and the predicted label often describe well the input image despite being wrong w.r.t the dataset label. For instance, the two failure cases in Fig. 7 select "stadium" and "dumpsite" when the ground-truth class names are "football" and "garbage site". In addition, for the first case, the exact name "football" is still available among the best candidate names, but our method considers "stadium" as a better fit. Another example is the last failure case in Fig 5, where the model assigns the name "nokia" to a Nokia 3310 cellphone, while the ground-truth class name is "cellphone". Also in this case, the ground-truth label is present in the candidate list but our method considers "nokia" a more fitting class.

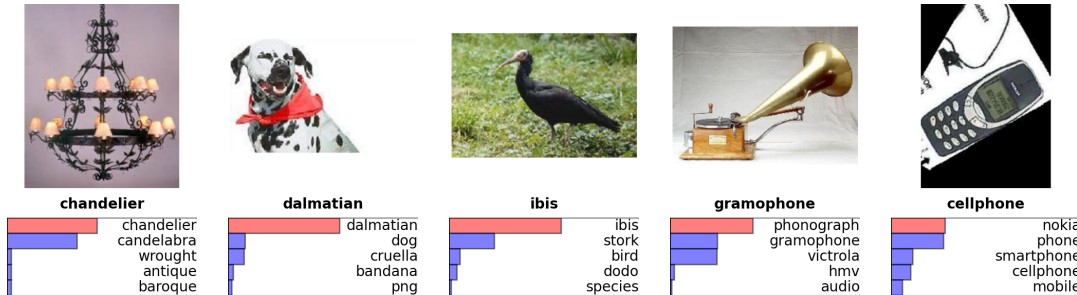

Figure 5: Qualitative results on Caltech-101. The first three samples represent success cases, the last two shows failure cases.

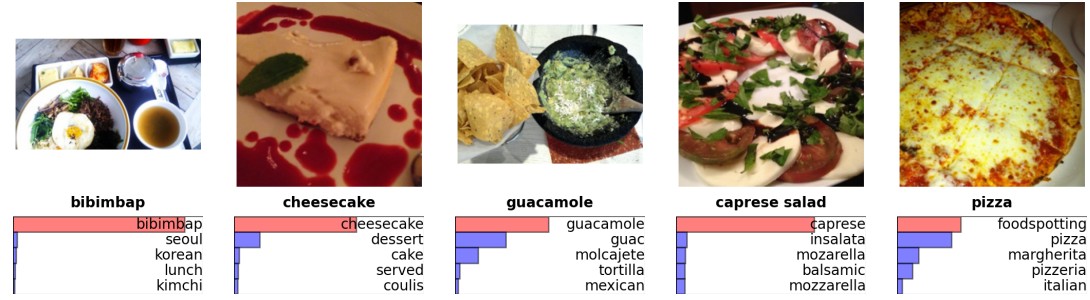

Figure 6: Qualitative results on Food101. The first three samples represent success cases, the last two shows failure cases.

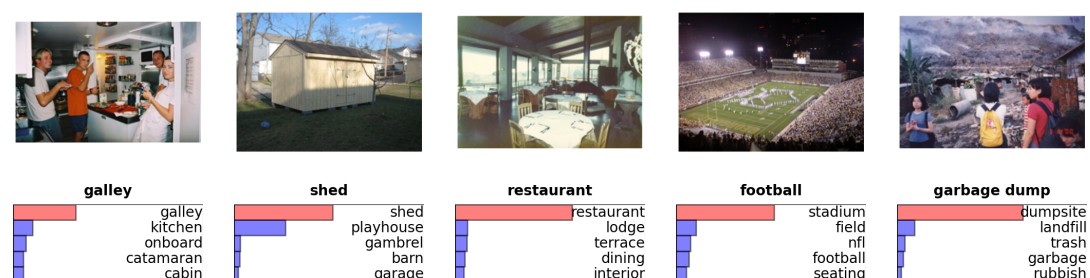

Figure 7: Qualitative results on SUN397. The first three samples represent success cases, the last two shows failure cases.

Finally, we notice the discovery of correlations between terms in the reasoning of our model. In the provided examples, it happens multiple times that the candidate class names do not describe objects in the scene but rather a correlated concept to the image. For instance, the third example in Fig 5 shows a Dalmatian, and among the candidate names there is "cruella", which is the name of the villain of the movie "The Hundred and One Dalmatians". Another instance of this appears in the first example of Fig 6, where the model correctly associates the "bibimbap" dish to its place of origin, Korea, with the candidate name "korean".

