# OpenReview forum: "Vocabulary-free Image Classification"
_NeurIPS.cc/2023/Conference — NeurIPS 2023 poster_

### Official Review · Reviewer_3fhc · 2023-06-29

**Soundness:** 3 good
**Presentation:** 3 good
**Contribution:** 2 fair
**Rating:** 5
**Confidence:** 4

**Summary:**

This paper proposes the task of vocabulary-free image classification -  image classification without a predefined set of categories (which is required in zero-shot classification). The paper further proposes the first method for this task - Category Search from External Databases (CaSED). For each given image, it first retrieves captions from an external text corpus, e.g. a large-scale caption dataset. The category names for the classification task are extracted from the retrieved captions via text parsing and filtering. For the classification, a scores fusion is conducted between 1) image-to-text matching score between the input image and the category names, and 2) the text-to-text matching score between the average embedding of retrieved captions and the embeddings of category names.
The technical contributions are
1) The paper proposes the task of vocabulary-free image (VIC), to overcome the constraint of requirement of a predefined set of class names
2) The paper proposes an approach for VIC task via extracting category names from an external large-scale text corpus

**Strengths:**

1) The paper is clearly written and easy to follow
2) Vocabulary-free image classification is an interesting task as it requires no prior knowledge of semantic class space of visual inputs.

**Weaknesses:**

1) Some details of baselines and evaluation metric are missing. See questions.
2) The choice of the caption database (as the text corpus) might have a big impact on the performance. Guidances / criteria of selecting a good text database are missing.
3) A baseline is missing: WordNet and English words contain different semantic words than words covered by captions in the subset of PMD. It would be interesting to extract the words from the captions in the subset PMD, and apply CLIP for image-text matching.
4) In Table 4(b) using YFCC100M (29.9M) as the text corpus leads to worse results than the case of CC12M (10.3M) and Redcaps (7.9M). Using CC12M alone leads to better results than using all the five datasets (54.8M). Further explanations are required.
It would be interesting to report, besides the number of caption,   a) the number of objects / nouns covered in each database, and b) the relevance of captions in a database to the image dataset, e.g. by reporting the semantic similarities between captions and semantic labels of the image dataset,  semantic similarities between captions and the visual contents of the image dataset.

Minor:
1) Table 4(b):  asterisk meaning missing (I assume it means a subset of YFCC100M)

**Questions:**

1) In Sec 5.2 line 275, the paper mentions that “applying CLIP on top of a pre-defined large vocabulary is not effective” based on the inferior performance on WordNet (117k words) and the English Words (234k words). However, as the first step, CaSED uses CLIP to retrieve K closest captions to an input image from a large-scale text corpus (54.8M captions). Why is it effective here?
2) For the cluster accuracy (line 242), is it necessary to make sure that the number of predicted labels (number of clusters) should be equal to the number of ground truth labels? How to make sure? What happens if these two are not equal?
3) For the baselines of BLIP-2 captioning (line 253), each image has one description. How to compute the semantic relevance and the cluster accuracy? How to determine the predicted label from this description? Is the description directly treated as a predicted label?

**Limitations:**

One missing limitation is that the method requires selection of a text database that is semantically relevant to the contents of the query image dataset. The choice of text corpus might have a big impact on the performance.

---

> ### Author Rebuttal · Authors · 2023-08-09
>
> **[W2: How to select databases]** The choice of the database is indeed crucial for CaSED, as shown in Tab. 6 of the Supplementary. We also perform additional experiments and calculate the average semantic similarity between the closest caption retrieved and the ground-truth for all the images in the dataset, reporting the results in Tab. A3 of the rebuttal PDF. This metric correlates well with the database performance across the three metrics. From the tables, we can see that:
> - The quality is more important than the dimension of the database (e.g., YFCC100M subset with 179.6K concepts performs worse than CC12M with 36.7K).
> - Datasets scraped from the web (e.g. CC12M) perform better than specifically annotated ones for captioning/VQA (e.g. Visual Genome). This might be due to the type of the descriptions, more general in the first case.
> Despite these observations, it is hard to find metrics to pre-compute the best database to use in the absence of test labels information. We will discuss this limitation in Sec. 6.
>
> **[W3: PMD Words baseline]** We extracted all the unique words from our PMD subset (around 2.25M words) and used them for classification with CLIP. We tested this new baseline on the ten datasets considered in the paper and we report the results on Tab. A1 of the rebuttal . Compared to CaSED, We achieve 14.2% cluster accuracy (-28.9%), 1.4% semantic IOU (-16.3%), and 27.4 semantic similarity (-23.0).
> This new baseline achieves the worst performance due to the extreme noise contained in web-scale databases, e.g., typos, neologisms, etc., and the usage of words instead of captions. Our analysis confirms that CLIP achieves suboptimal performance when applied to a list of class names rather than captions, a phenomenon accentuated when the list is uncurated.
>
> **[W4.1: Objects per database]** We extracted all the unique words from the databases and performed POS tagging. We report in Tab. A3 of the rebuttal PDF the statistics for nouns, adjectives and verbs, removing all the other tags. From the analysis of the POS tag distribution, it is hard to extract any pattern that justifies the performance of different databases. For instance, CC3M is the only database without an extreme imbalance for nouns (i.e., 57.9% compared to >96% for the other in the top-5) while still achieving good performance. Moreover, while the number of concepts depends on the size of the database, there is no clear correlation between this value and the achieved scores. As an example, Localized Narratives has a comparable amount of concepts w.r.t. CC3M but a gap of >20% in cluster accuracy, and of ~10 points in semantic similarity and IoU.
>
> **[W4.2: Relevance of captions]** We additionally evaluated the relevance of the captions w.r.t. the class names of ImageNet, the dataset we used for hyperparameter tuning. We report the results in Tab. A3 of the rebuttal PDF. For each image, we retrieve the closest caption in the database and use it to evaluate the semantic similarity with the ground-truth label. We call this metric Closest Caption Semantic Similarity (C.C. S-Sim). Interestingly, the best performing database achieves the highest C.C S-Sim, suggesting that a higher similarity score may correlate with the effectiveness of a database. However, although our custom database achieves the same C.C. S-Sim score as CC12M (i.e., the best database from our analyses), our split still falls behind on the three metrics, suggesting that this score may not be sufficient to assess the quality/relevance of a textual database.
>
> **[W5: YFCC100M asterisk]** Thanks for spotting this. Indeed, we use the subset of YFCC100M, used for PMD (L261 and L305). We will state it clearly in the final manuscript.
>
> **[Q1: Effectiveness of retrieval]** CLIP was trained to match images with their respective captions, therefore captions better reflect the training input distribution used for CLIP. Using only class names is suboptimal [a], hence the flourishing field of prompt engineering/tuning for CLIP [b].
> Moreover, from our preliminary study (Fig. 2 of the main paper), we discovered that captions hold a strong classification signal. When comparing the captions generated by a captioning model (i.e., BLIP-2) with those retrieved from an external source of information, the latter provide a stronger signal, thus justifying our rationale of using retrieved captions for candidate generation.
> [a] Radford, et al. "Learning transferable visual models from natural language supervision." ICML, 2021.
> [b] Zhou, et al. "Learning to prompt for vision-language models." IJCV 2022.
>
> **[Q2: Details on cluster accuracy]** There is no requirement for a one-to-one match between the number of predicted clusters and the number of ground-truth clusters. Please, see the global response for further details.
>
> **[Q3: BLIP-2 captioning]** We ask BLIP-2 to generate a caption given the input image and we directly treat such caption as the predicted label. Interestingly, we noticed that the captions are often repeated for similar input images. Indeed, on average across the ten datasets, BLIP-2 Caption generates 3000 different captions, clustering samples into a finite number of groups. Since we have a number of clusters smaller than the number of images, we can evaluate the cluster accuracy using the captions and the ground-truth labels. For the semantic similarity and IoU, we follow the same procedure we used for the other approaches, evaluating them between the predicted label (i.e., the caption) and the ground-truth class.
>
> **[L1: Database relevance to query]** We expect that testing CaSED using a generic database on a specific dataset (e.g., medical data) might result in bad performance. However, CaSED can be enriched with domain-specific textual information (e.g, medical corpora) without training, making it much more flexible and applicable w.r.t. a fine-tuned model. We will add these considerations in the final version.

---

> > ### Comment · Reviewer_3fhc · 2023-08-17
> > **Rating changed into borderline accept.**
> >
> > I appreciate the efforts of the authors and feel that most of issues are addressed (more or less). However, I am still concerned about the limitation of selection of a proper text corpus which has a big impact on the performance, as there is no clear unsupervised criterion guiding in this regard.
> > Overall, I change my rating into borderline accept.

---

> > > ### Author Response · Authors · 2023-08-18
> > >
> > > We thank Reviewer 3fhc for their valuable comment and for their willingness to increase their score.
> > >
> > > We absolutely agree that the choice of the textual database has an impact on the performance. However, in the paper, we show that choosing a large database (e.g. a subset of PMD) provides good results across 10 different benchmarks with totally different visual domains, e.g. from cars (Stanford Cars) to textures (DTD), from satellite images (EuroSAT) to food (Food-101). These results are achieved without any tuning, selection, or pre-filtering of the textual database, hinting that the use of a large corpus can already suffice for a variety of applications.
> > >
> > > Interestingly, we found that the average performance on the 10 datasets of a chosen textual database generally correlates with the accuracy on ImageNet (Tab. 6 of the supplementary) and with the closest caption semantic similarity (C.C. S-Sim, Tab. A3 on the rebuttal PDF). Examples are the subset of PMD and CC12M, achieving the best average results on the 10 datasets according to all metrics (Tab. 4b of the main paper) while being the best on the ImageNet validation set w.r.t. both the closest caption semantic similarity and the ViC metrics.  Note that, to calculate the closest caption similarity and the performance on ImageNet we do not need to access the test set for the dataset at hand. The ImageNet validation set can thus be used as a proxy to pick the textual database in case of a lack of priors on the test set.
> > >
> > > While these results are promising, the 10 datasets cannot encompass all possible use cases. Thus, we will expand on this limitation in Sec. 5,  highlighting how a proper automatic selection/extension of the textual database depending on the test samples is an interesting direction for future work.
> > >
> > > We thank the reviewer for acknowledging the effort we put in this rebuttal and for the increased rating. However, just to double check, we do not see the change of rating in the original review and would like to confirm that the reviewer has indeed changed it in the system.

---

### Official Review · Reviewer_xx1r · 2023-07-07

**Soundness:** 3 good
**Presentation:** 3 good
**Contribution:** 2 fair
**Rating:** 3
**Confidence:** 5

**Summary:**

This work introduces Vocabulary-free Image Classification (VIC), a task that aims to assign an image to a class in a large, evolving semantic space without a known vocabulary. The proposed method, Category Search from External Databases (CaSED), utilizes a vision-language model and an external database to extract candidate categories and assign the best matching category to the image. Experimental results demonstrate the superiority of CaSED over other frameworks, offering efficiency and promising future research opportunities.

**Strengths:**

The strengths of this work can be summarized as follows:
1.	Exploration of Vocabulary-free Image Classification (VIC) as a task, which overcomes the limitations of existing Vision-Language Models (VLM) for image classification.
2.	Proposal of CaSED, a training-free method for VIC that utilizes large captioning databases. CaSED does not require additional parameter tuning or fine-tuning of textual and visual encoders.
3.	Consistent outperformance of CaSED over a more complex VLM, BLIP-2, on VIC. CaSED achieves superior performance while utilizing significantly fewer parameters.
4.	Introduction of specific evaluation metrics for VIC, providing a valuable reference for future research and benchmarking in this domain.


**Weaknesses:**

1.	The most interesting part of this paper is the new task: vocabulary-free image classification. However, it is already studied in the existing work: https://openreview.net/forum?id=sQ0TzsZTUn [1]. The major differences between this paper and the existing work are as follows:
a)	The authors of the current paper extend the evaluation by using a larger vocabulary set -- BabelNet, compared to the existing work.
b)	The authors adapt the text-to-text score to find the best matching class.
From the perspective of the reviewer, these two differences may not be considered major advancements. Further analysis and comparison with the existing work are required to fully understand the extent of novelty and contributions of the current paper.
2.	To enhance the clarity and completeness of the paper, it would be beneficial for the authors to provide additional information on how to calculate the classical Cluster Accuracy, particularly considering the open vocabulary nature of the predicted labels. Given that the vocabulary is unconstrained, it is important to explain how the accuracy metric accounts for potential variations in the predicted labels.
3.	It is worth considering that most of the class names in vocabulary-free image classification may consist solely of nouns. Consequently, retrieving information from image captions, which typically contain more diverse linguistic elements, could be inefficient and computationally expensive. This raises a valid concern about the practicality of such an approach, as it may not yield meaningful results and could potentially waste computational resources. The authors should address this issue by discussing alternative strategies or potential optimizations to mitigate the computational burden associated with retrieving information from image captions.

[1] Han, Kai, et al. "What's in a Name? Beyond Class Indices for Image Recognition." arXiv preprint arXiv:2304.02364 (2023).


**Questions:**

Please address my comments in the weakness section.

---

> ### Author Rebuttal · Authors · 2023-08-09
>
> **[W1: Relationship with [1]]** The cited paper was released on arXiv concurrently (April 5th) with the deadline of NeurIPS (May 17th), making it “contemporaneous” according to the NeurIPS call for papers. The provided OpenReview link leads us to a withdrawn paper from ICLR 2023, with unanimously “below acceptance threshold” scores. Nevertheless, our paper differs on two macro points.
>
> First, while the arXiv paper proposes a classification approach based on a fixed vocabulary such as WordNet, our paper utilizes a different methodology by automatically generating the vocabulary using an external database of textual descriptions. In this regard, we would like to stress that we are not using BabelNet, which we mentioned in the manuscript only for statistical purposes. This distinction allows our approach to handle the task in a unique manner that is different from [1] which requires a pre-defined well-curated vocabulary as WordNet.
>
> Additionally, the arXiv paper relies on accessing all the images beforehand, performing classification and refinement offline. On the other hand, our approach generates the vocabulary independently among samples, working in an online fashion. Our database can be updated online without human supervision, while [1] needs human annotations to introduce new concepts.
>
> Taking these points into consideration, we firmly believe that our work tackles a more general setting than [1]. We are confident that these differences make our contribution valuable and provide significant insights to the field. While we cannot provide a fair comparison with [1], as there is no public implementation available, we are open to including this discussion on the differences in Sec. 2.
>
>
> **[W2: Details on cluster accuracy.]** Please, see the global response.
>
>
> **[W3.1: Captions may not yield meaningful results]** Indeed, captions contain more than just mere class names. However, this is their strength in the context of this task and enables superior performance of our pipeline. CLIP is trained to match images with their respective captions, therefore captions are the form of input that better reflect its training distribution. Using only class names is well known to be suboptimal [a], hence the flourishing field of prompt engineering/tuning for CLIP [b,c]. This is also confirmed by our results in Tab. A1 of the rebuttal PDF. Here, we test CLIP with a vocabulary created from all the words defined in our text database, showing that using the complete list of words is not an effective strategy, achieving the worst results among all baselines.
>
>
> **[W3.2: Computational cost]** From a computational standpoint, captions are only encoded once for creating the faiss index [d], therefore the overhead is negligible. Moreover, storing the embedding of a class name or of a caption has the same memory cost. Finally, as we show in Tab. 2 of the Supplementary, the inference time of our method is still low compared to, e.g., caption generation or VQA. Indeed, our method takes about 4370 ms to classify 64 images on average, while BLIP-2 takes 5670 ms to generate the response. Last, the dimension of the knowledge base does not imply any time overhead, as we achieve the same inference times when we use our database of 55M captions or LAION-400 with its 414M captions.
>
> [a] Radford, Alec, et al. "Learning transferable visual models from natural language supervision" ICML 2021.\
> [b] Zhou, Kaiyang, et al. "Learning to prompt for vision-language models" IJCV 2022.\
> [c] Zhou, Kaiyang, et al. "Conditional prompt learning for vision-language models" CVPR. 2022.\
> [d] Johnson, Jeff, Matthijs Douze, and Hervé Jégou. "Billion-scale similarity search with gpus" Transactions on Big Data 2019.

---

> > ### Comment · Reviewer_xx1r · 2023-08-20
> > **Thanks for the response and still concern about the novelty.**
> >
> > I appreciate the response on my concerns. The difference with the other paper is clearer to me now. First, I agree that compared with the paper I pointed out, this submission leverages an external text caption database, while the other work uses an unconstrained dictionary. This is slightly different. Second, the statement that “our approach generates the vocabulary independently among samples, working in an online fashion” is not true. The CaSED method is not generating any vocabulary independently. Instead, a predefined and fixed large-scale text database is given, and for each image, the large-scale text database needs to be fully scanned to retrieve the relevant captions. Hence, no independent vocabulary generation is involved. It is retrieval from the extra text database.
> >
> > After reading all reviewers comments and the rebuttal response, I remain rather concerned about this submission.
> >
> > 1. The novelty of the paper is rather limited.
> >     a. From the method perspective, the use of VLM for classification is not new, leaving the main pitch of this paper to be the use of an external text database to retrieve the most “relevant” captions, and then assign a category name based on the retrieved captions. This is a minor change on top of the CLIP model. Meanwhile, using external text database to retrieve helpful information is a very common method in NLP, e.g.[a], as also mentioned in the related work. However, this idea has also been well studied in the computer vision literature, aside from those mentioned in the submission, there are other ones that are more relevant, e.g., using an external text database to facilitate image recognition [b], image captioning [c], generalized class discovery [d], among others.
> > [a] Lewis,Patrick,  et al, Retrieval-Augmented Generation for Knowledge-Intensive NLP Tasks, NeurIPS 2020
> > [b] Choudhury, Subhabrata, et al, The Curious Layperson: Fine-Grained Image Recognition without Expert Labels, BMVC 2021
> > [c] Ramos, Rita,  et al, Retrieval-augmented Image Captioning, EACL 2023
> > [d] Ouldnoughi, Rabah, CLIP-GCD: Simple Language Guided Generalized Category Discovery, arXiv:2305.10420
> >  &nbsp;
> >     b. From the task perspective, the so-called “Vocabulary-free Image Classification” (VIC) is not  “vocabulary-free”, as pointed by other fellow reviewers, due to the fact that the very large vocabulary, in the form of a large-scale caption dataset,  is required. It says the “The objective of VIC is to assign an image to a class that belongs to an unconstrained language-induced semantic space at test time, without a vocabulary”. This is not true. The used caption text dataset is a superset of the vocabulary, which contains more abundant information that simply an open-vocabulary.
> >     &nbsp;
> >     Indeed, it appears to me that the VIC is essentially the open-vocabulary recognition, for which there are plenty of studies already, with methods addressing even harder problems of object detection (i.e., box level image recognition) and segmentation context (i.e., pixel level recognition). How does VIC differ from them?  The whole literature along this direction was omitted.
> >     &nbsp;
> >     For example,
> >     Ghiasi, Golnaz, et al, Scaling Open-Vocabulary Image Segmentation with Image-Level Labels, ECCV 2022
> >     Zareian, Alireza, et al, Open-Vocabulary Object Detection Using Captions, CVPR 2021
> > &nbsp;
> >
> > 2. The choice of the text database is very crucial. There is no guarantee that the text database can cover the semantic names of the testing images. This can not be reliably used in practice. On the other hand, if the semantic names are known beforehand, the problem becomes a close-set classification problem, contradicting its motivation. In this sense, there is no guarantee that the method can work or not in praticse.
> > &nbsp;
> > 3. The inference time is as slow as the generative methods, e.g., BLIP-2, which is handling a more difficult problem, by directly generating the caption containing the name for the given images. For the comparison with BLIP-2, which was questioned by other reviewers as well, I also think it may not be fair. The proposed method is given a database to “look up” the category names, but the BLIP-2 merely leverages the input image to directly predict the category names.
> >
> > Therefore, the novelty is deeply disconcerting from both the standpoint of the task at hand and the methodology employed. In light of the comprehensive examination of comments and discussions, my apprehensions have been significantly heightened.
> >
> > Hence,I would keep my initial rating.

---

> > > ### Author Response · Authors · 2023-08-21
> > >
> > > **[Differences with [1]]** We are glad that the reviewer acknowledges the differences between the two settings. However, we believe that some crucial aspects still need clarifications. First, [1] proposes a classification approach based on a **fixed and curated** vocabulary of words (such as WordNet), while we use unstructured and non-curated data, and propose a new methodology to automatically extract the vocabulary (i.e. set of candidate classes) on the fly. This is different both from a problem formulation standpoint (i.e. we need to create a set of candidate classes from all captions) and from a technical point of view, as we show that using mere names extracted from captions does not achieve the same results (Tab. A1 of the rebuttal, PMD Words baselines).
> > >
> > > Second, while [1] needs to have all images of the target dataset simultaneously available, we produce a set of candidate categories **independently** for each sample, i.e. performing instance-wise vocabulary prediction. As described in Sec. 4.1 of the main manuscript, we first collect the closest captions to the input image, and then extract a finite set of categories by performing text parsing and filtering. Since these two operations are independent for each input image, our method effectively generates a vocabulary (i.e. a set of target classes) independently among samples. This implies that CaSED can work online (i.e. one sample at the time, in a stream), while methods applied for [1] cannot.
> > >
> > >
> > > **[Novelty concerns on the method]** We do not claim that our novelty lies in using VLM for classification or introducing retrieval in vision. Both these statements do not consider that we are targeting a new task and that we fully tailored the retrieval (Sec 4.1) and scoring mechanism (Sec 4.2) to the wide search space of the proposed task.
> > > In addition, the works provided to support such claims [b, c, d] have a completely different nature. [b] performs unimodal retrieval, [c] uses retrieved captions as input to train a captioner, and [d] exploits the database for clustering. None of them revises the VLM classification pipeline using an external caption database without training, as we do. The most similar work to ours is [d], which solves the problem of image classification using a VLM equipped with retrieval. However, the work was published on arXiv the same day of the Paper Submission Deadline of NeurIPS, making it a concurrent work according to the NeurIPS Call for Papers guidelines. We are happy to cite [d] as concurrent work in Section 2.
> > >
> > >
> > > **[Novelty concerns on the task]** With the first sentence, we assume that the reviewer refers to a question raised by reviewer panm. However, their concern was about the terminology “vocabulary”, which they considered addressed after the rebuttal (also raising the score). Overall, no reviewer raised concerns about the task definition. Instead, three reviewers (m2jc, 3fhc, and xx1r) out of four listed the task definition and protocol among the strengths.
> > >
> > > The statement that *vocabulary-free is essentially open vocabulary* is incorrect. The term “open vocabulary” is used for detection and recognition tasks, and refers to the freedom in specifying any free-form text as input, but the vocabulary is given and usually composed of few candidate classes (e.g., 20 for Pascal VOC, 150 or 847 for Ade20k, 65 for COCO Objects as per the papers cited by the reviewer). This is substantially different to our setting where no pre-defined vocabulary is given and the semantic space is extremely large.
> > >
> > > As additional note, with the baseline required by reviewer 3fhc (i.e., the PMD Words baseline, Tab. A1) we showed that simply using all the words in the text database to perform image classification is not an effective approach for this task (i.e. 27.4 S.Sim. vs 50.4 of ours). This suggests that having an exhaustive list of words (potentially containing the ground-truth class names, i.e., a superset) is insufficient to solve the task. Moreover, the usage of a retrieval database is a design choice of our method, it has nothing to do with the task itself. One could theoretically solve the VIC task with a VQA and/or captioning model (like BLIP-2), although we showed that the current state-of-the-art captioning model does not yield strong performance for the task.

---

> > > > ### Author Response · Authors · 2023-08-21
> > > >
> > > > **[Importance of the text database]** We absolutely agree that the choice of the textual database has an impact on the performance. However, in the paper, we show that choosing a large database (e.g. a subset of PMD) provides good results across 10 different benchmarks with totally different visual domains, e.g. from cars (Stanford Cars) to textures (DTD), from satellite images (EuroSAT) to food (Food-101). These results are achieved without any tuning, selection, or pre-filtering of the textual database, hinting that the use of a large corpus can already suffice for a variety of applications. While this does not encompass all possible use cases, it still shows the viability of the approach in practical applications. Even for specific ones, one may just specify an appropriate database (or domain) and download related captions, without the need to manually provide a list of class names and thus working in the closed-set scenario.
> > > >
> > > > Finally, we would like to stress that CaSED is a first step to address the challenging ViC task. We acknowledge its limitations (that we will expand in Sec. 5 and 6) but we hope that, together with our benchmark, it can serve as reference for future research on this domain (as acknowledged as strength in xx1r’s review).
> > > >
> > > >
> > > > **[Unfair comparison with BLIP-2]** We would like to clarify that other reviewers requested more information on how to evaluate the cluster accuracy with captions and on the prompts. No reviewer raised concerns about the comparison with BLIP-2. Instead, all the other reviewers shared positive opinions on the task (m2jc, 3fhc, xx1r), the method (panm, xx1r), and the experiments/ablations (panm, m2jc, 3fhc).
> > > >
> > > > As reported to reviewer panm, BLIP-2 expands CLIP with an additional 129M samples for training the Q-former and a text-only decoder (OPT or T5) that are trained on trillions of tokens. In comparison, our work is training-free, and we require less data (i.e., our textual database is around 55M captions). Last, BLIP-2 was specifically trained to solve generation tasks, while our method is able to create a list of candidates without the need for finetuning or training. Since our method works with a database which is roughly a subset of the one used by BLIP-2, one could argue the opposite, i.e., on the contrary, BLIP-2 is unfair to our method.
> > > >
> > > > In addition, we are 1.3x faster than BLIP-2 (as reported in the supplementary material and in the rebuttal), and still outperform BLIP-2 with the same architecture on average across the ten datasets across the three metrics considered for the task, as the reviewer recognised among the strengths of the method. As a side note, we want to highlight that retrieval is performed on CPU, and it can be massively sped-up by using GPUs (which are supported by Faiss).
> > > >
> > > > We do not claim our task to be harder or our method to be better than BLIP-2 on all possible tasks. We tackle a well-defined task and propose a solution that is more efficient and accurate than BLIP-2.
> > > >
> > > >
> > > > **[Final score]** While we totally respect the opinions and criticisms, we would like to point out that there is a misalignment between xx1r’s review and the score. According to the guidelines, a score of 3/Reject is reserved “*For instance, a paper with technical flaws, weak evaluation, inadequate reproducibility and incompletely addressed ethical considerations*”. However:
> > > > The review considers the paper to be sound (score of 3 out of 4, the same given by all reviewers), something that contrasts with the presence of technical flaws. From the comments, the reviewer seems mostly concerned with “novelty”, which has nothing to do with technical flaws.
> > > > The reviewer states in the strengths that  “*consistent outperformance of CaSED over a more complex VLM, BLIP-2, on VIC. CaSED achieves superior performance while utilizing significantly fewer parameters*” and that our proposed benchmarks and metrics provide “**valuable reference for future research and benchmarking in this domain.**” This contrasts with criticisms on weak evaluation
> > > > There are no mentioned reproducibility issues in the above review, and the ones raised by m2jc were some missing details that we clarified.
> > > > None of the reviewers posed ethical concerns (“no ethics review needed”)
> > > > Finally, all our technical contributions (i.e. the task and method) are listed as “Strengths” in their review, mentioning “promising future research opportunities” in the summary of the work.
> > > > Given these considerations, we believe the score of 3 (and even more the 2 at the end of xx1r’s review) does not reflect the quality and potential impact of the paper as acknowledged by Reviewer xx1r themselves.

---

### Official Review · Reviewer_m2jc · 2023-07-12

**Soundness:** 3 good
**Presentation:** 2 fair
**Contribution:** 3 good
**Rating:** 7
**Confidence:** 3

**Summary:**

Current approaches to zero shot classification are limited by the assumption of an existing set of candidate classes. This paper presents a novel task, Vocabulary-free Image Classification, where semantic categories need to be automatically mined. This is important when generalising to a new domain, or the domain evolves over time. To address this task, the paper proposes to search for categories from an external database. While conceptually simple, this proves to be an effective approach, and performs better than using captioning models or large lexical databases like  WordNet. The paper also proposes metrics to evaluate on this new problem.

**Strengths:**

- The paper addresses an important limitation of prior zero-shot classification approaches.
- The importance of the proposed task, the motivation behind the method, and the method itself, are clearly laid out.
- The components of the method are well ablated
- Multiple baselines are proposed and analysed
- It presents an avenue for future research on how to discover semantic concepts in the V+L space

**Weaknesses:**

1. Text-to-text score. Using CLIP as a scoring function for image-to-text makes sense. However, it is not obvious that a CLIP text encoder is optimal for the text-to-text score. L207-208 say that the CLIP text encoder ”focuses on the visual elements of the caption”. This could be backed by showing inferior performance when using a text-only model for the text-to-text score. Furthermore, an addition to the semantic space representation experiments (L136-170) would be a model that uses CLIP to retrieve captions, and then a text-only encoder to match the captions to a  semantic class.

2. Reproducibility. The paper does not mention how BLIP was prompted to generate class names

3. Table 4a. The numbers on the last row of table 4a (retrieval with vis + lang scoring) correspond to the avg. values on Tables 1, 2 and 3. However, the first row (BLIP2) does not correspond to any row on these tables. (Related to 2. Reproducibility -- it not clear how the different settings differ)

4. Multimodal scoring results. Table 4 a) shows that scoring with the text-to-text score performs better than image-to-text score, and adding the image-to-text contributes a small amount. This means that the average caption is the most informative signal. That makes the statement on L192-193 "The visual information is the most reliable source for scoring the candidate categories" invalid. It is indeed an interesting finding and some intuition behind that would be useful. It would be interesting to see the performance of an ever simpler baseline that only uses text - select the class that has the highest number of occurrences in the retrieved captions.

5. Candidate classes. What is the distribution of the number of extracted candidate classes? And how does it differ as you change the number of retrieved captions?

6. Generalization: One of the motivations behind this work is the applicability of ZS classifiers to domains that might not have a closed set of classes available. However, the current approach relies entirely on datasets of image-text pairs, from which captions are mined. For a domain that is not sufficiently represented in such datasets (e.g. medical images), the proposed method will fail. A discussion of how text-only databases can be used to handle that would be nice to see.

7. Database. In Table 4b you show that simply using the CC12M set of captions performs better than the contracted one (Ours). Why not use that?

**Questions:**

1. Cluster Accuracy. When measuring cluster accuracy, one can imagine that in the vocabulary free case, there might be many more predicted labels that in the ground truth (up to the number of samples of the dataset). How is that handled?

+ Please see weaknesses

**Limitations:**

Then limitations look good. A limitation not discussed is Weaknesses 6. Generalization

---

> ### Author Rebuttal · Authors · 2023-08-09
>
> **[W1: Text-to-text score]** When performing text-to-text scoring with a text-only model, Sentence-BERT [a], and considering only text scores (i.e., alpha=0), we get 39.8% cluster accuracy (-2.9%), 51.9 semantic similarity (+1.6), and 16.6% semantic IoU (-0.4%) w.r.t. CLIP. Thus, CLIP is better at sample grouping and slightly better at the overlap between predicted and ground-truth labels. The semantic similarity is instead better captured by Sentence-BERT: this is not surprising as the latter is also used to compute this metric (L238). The two models are on par when performing multimodal scoring (e.g., +0.5% of CLIP on cluster accuracy while -0.2% on semantic IoU). These results support the use of CLIP to perform text-to-text matching. Note that, nevertheless, we must use a VLM text encoder to retrieve captions in the first place. Thus, it is not possible to fully replace the CLIP text encoder with a text-only model. Thus, using the latter only for the multimodal scoring will lead to an increase in the total number of parameters.
>
> [a] Reimers and Gurevych. "Sentence-bert: Sentence embeddings using siamese bert-networks" EMNLP-IJCNLP 2019.
>
>
> **[W2: Details on BLIP-2 prompts]** We followed the BLIP-2 demo for prompting. Specifically, for captioning, we used the prompt “Question: what's in the image? Answer:”. For VQA, we used “Question: what's the name of the object in the image? Answer: a”. We will clarify the prompting in the final manuscript.
>
>
> **[W3: BLIP-2 in Tab. 4]** Tab. 4a ablates how different candidate generation strategies, i.e. generative v.s. retrieval, impact our method performance. In this ablation, we exploit BLIP-2 as the generative module for candidate generation by prompting “Question: list the objects in the image. Answer:”. This generates a list of comma-separated objects that we then parse. This is different from Tab. 1-3, where BLIP-2 is either a captioner or a VQA system. For better clarity, we will replace the name “BLIP-2“ with “Generative” (in contrast to “Retrieval” on the second row) in Tab. 4a to highlight how we generate candidates.
>
>
> **[W4.1: Reliability of image vs text]** We will remove the confusing comment and clarify what we mean. In Tab. 4a we reported three configurations of CaSED, where the best uses both the image-to-text and the text-to-text scores. However, as mentioned in the implementation details, the best value of alpha (i.e., the weight to average the multimodal scores) is 0.7, i.e., 0.7 * image-to-text-score + 0.3 * text-to-text-score, effectively using more visual than textual information. However, from the direct comparison of alpha=0.0 (text-to-text only) and alpha=1.0 (image-to-text only) it is evident that, when using a single modality, it is best to use the caption centroid. This counterintuitive behavior might be due to the modality gap, which makes it harder to linearly interpolate between the two modalities. Finally, note that the captions are retrieved in the aligned feature space using image representations as a query, and therefore they directly depend on the input image.
>
>
> **[W4.2: Occurrences baseline]** As suggested, we modified CaSED to predict semantic names as the most occurring word (CaSED top-1) instead of using multimodal candidate scoring. In case we encounter words with the same occurrences, the method uses our multimodal scoring for prediction. We test this baseline on the ten datasets and report the results in Tab. A1 of the rebuttal PDF. W.r.t our method, CaSED top-1 achieves 36.9% (-6.2%) for cluster accuracy, 14.4% (-3.2%) for semantic IoU and 47.4 (-3.0) for semantic similarity. Thus, simply selecting the most occurring word in the retrieved captions performs worse than scoring the words with our multimodal procedure. Such downgrade may be explained by the lack of any mechanism that considers the semantic similarity between modalities. Indeed, the cosine similarity used by CLIP and our multimodal scoring can effectively filter out terms that are less indicative of the subject in the image.
>
>
> **[W5: Distribution of candidates]** In Tab. A2 of the rebuttal PDF, we report the number of unique candidates extracted by the candidate filtering procedure, averaged over the ten datasets, and with an increasing number of retrieved captions, i.e., 1, 2, 5, 10, and 20. In the table, we show both the number of candidates extracted and the number of selected words. As the number of retrieved captions increases, the unique number of candidate words also increases, i.e., from 3849 with 1 caption to 28781 with 20. However, the number of selected words stabilizes around 800 as soon as we retrieve more than 1 caption. Having more captions reduces the noises in the selected words that might be present in a single caption.
>
>
> **[W6: Generalization]** The databases in our method are as relevant as the training dataset is for trained models. Consequently, it is challenging to retrieve concepts that are not well represented in the database, as it is hard to classify such concepts for trained models. CaSED, as a training-free method, can flexibly mitigate this issue by including new concepts in the database, without any retraining. Additionally, as the reviewer pointed out, our method is not restricted to the use of image-text databases and it is fully compatible with text-only datasets. We will discuss this limitation in the final manuscript.
>
>
> **[W7: Use CC12M]** We opted to use a mixture of web-scale databases (a subset of PMD), instead of cherry-picking the best performing database on ImageNet (CC12M) because we wanted to foster source diversity among databases. This is motivated by the idea that a user could simply download captions from the web without filtering or prior knowledge on the performance to solve the VIC task.
>
>
> **[Q1: Cluster accuracy]** Please, see global response.

---

### Official Review · Reviewer_panm · 2023-07-14

**Soundness:** 3 good
**Presentation:** 4 excellent
**Contribution:** 2 fair
**Rating:** 7
**Confidence:** 4

**Summary:**

The paper presents a method to perform image classification by using the similarity text and image embeddings of a large vision-language model (VLM). The method is called "vocabulary-free" because the large vision-language model doesn't have a particular set of classes (vocabulary) in mind, and (contrary to zero-shot approaches), the user does not need to provide a list of prompts. In particular, a set of k closest captions to the image embedding is retrieved from the VLM, and this candidate set is later scored to predict the image class. The underlying VLM is a ViT-L CLIP model.

**Strengths:**

- The paper shows very good results when compared with different reasonable baselines (see Table 1, in particular). The experiments use several popular image classification datasets, and use different metrics, and the proposed method is significantly better than the baselines in virtually all cases, and certainly on average.
- The figures in the paper help understanding the proposed approach (Figure 3) in detail and the difference between "vocabulary-free" and zero-shot image classification.
- The paper contains results from ablation studies used to justify certain design choices (e.g. number of candidate captions retrieved from the database), the importance of the visual and language scores, etc (see section 5.3). I found these experiments very interesting (modulo some questions that were unanswered, see later comments).

**Weaknesses:**

- The term "vocabulary-free" is a bit misleading, since the VLM does have a vocabulary built-in, i.e. the so-called "semantic space". It's impossible to classify an image representing an object outside this semantic space. However, it's true that the semantice space is far larger than the class space of any of the datasets used for evaluation (although not necessarily a superset). What the authors mean by "vocabulary-free" is precisely defined in the paper (section 3), so this weakness is not critical, but as mentioned before, I find the term not accurate.
-  My main concern with the paper is regarding the use of the datasets used to train the VLM model used by the proposed method. Section 5.1 mentions that the nearest-neighbours are searched over a collection of 5 large datasets, and the authors ablate the impact of each dataset in section 5.3 (table 4b). However, it's unclear the impact of the dataset used to train the VLM when compared with the baselines. To be more specific: is the proposed CaSED method than BLIP-2 VQA because the dataset used to perform knn is bigger, because the embeddings are better, or because knn+scoring is better than VQA (regardless of the dataset)?
- Lines 164-165 refer to the number of parameters as a reason for better/worse speed. This is technically wrong. The number of parameters can be (extremely highly) correlated with the speed, but it's not a direct cause of it necessarily. Thus, if arguments about speed are to be made, real runtime/query (on a given hardware) or -at the very least- FLOP/query (amortized over a given batch size) should be given instead.

---------
Update after rebuttal: The authors have (quite successfully) addressed my comments in their rebuttal. Thus, I'm slightly increasing my score.

**Questions:**

- Have you tried with other strategies to compute the final candidate scores? For instance, why not a log-linear average between the text-to-text and image-to-text scores?

**Limitations:**

The authors have adequately adressed the limitations and broader impact of the paper in section 6.

---

> ### Author Rebuttal · Authors · 2023-08-09
>
> **[W1: VLM built-in vocabulary]** Strictly speaking, VLMs, such as CLIP, process sequence of tokens. Thus, they have a possibly infinite vocabulary induced by all the possible combinations of tokens, i.e., the tokenizer can process any string/class name. While there is no performance guarantee for unseen classes, CLIP shows decent extrapolation performance [a]. Therefore, we exploit these properties to avoid using a fixed pre-defined vocabulary. We propose to use large internet-sourced databases, that we assume contain all the concepts available at a certain time with the possibility of further inclusion of new concepts as it evolves. This is better than existing approaches as they cannot recognize classes outside the fixed human-annotated vocabulary, and require additional human effort to update when new concepts appear.
>
> Regarding the use of ”-free” in the name of the task, we reckon the connotation is within the common naming practice in computer vision. One clear example is the accepted term “task-free continual learning” [b]. “-free” in this context does not mean the non-existence of tasks, but rather that task boundaries are not given to the model during training and inference, and the model should infer them on the fly. This is consistent with our definition of “vocabulary-free”, as the vocabulary for each classification task is not given apriori, it is rather inferred (or generated) through our candidate proposal routine. However, we understand that confusion might arise from the term, so we are open to improving the naming in case of suggestions.
>
> [a] Radford, et al. "Learning transferable visual models from natural language supervision" ICML 2021.\
> [b] Aljundi, Rahaf, Klaas Kelchtermans, and Tinne Tuytelaars. "Task-free continual learning" CVPR 2019.
>
>
> **[W2: Why is CaSED better than VQA?]** We understand the concerns of the reviewer. However, we believe that our comparisons are fair in terms of data and that the improvement is due to the novel methodology we present. Here are the reasons for this statement:
> - We use the original CLIP model trained on 400M image-text pairs and in addition we retrieve from a 55M text-only samples database (PMD). In comparison, BLIP-2 uses the same CLIP model trained on 400M samples, an additional 129M dataset (roughly a superset of PMD) for training the Q-former and a text-only decoder (OPT or T5) that are trained on trillions of tokens. Therefore it seems clear that our data is roughly a subset of the data used by the strongest baseline (fairness of simpler baselines are described below).
> - All the baselines (including BLIP-2) use the same visual embeddings: CLIP embeddings. Therefore the improvement cannot come from different embeddings.
> - To further investigate if the data is the driver of the improved performance, for this rebuttal we also drafted a new baseline, PMD Words, where we extract a vocabulary composed of all the semantic terms contained in the database that we consider (PMD 55M). This elucidates whether the improvement comes from the lack of concepts in other baselines or from the novel methodology. We report the result in Tab. A1 in the rebuttal PDF. We discover that this baseline produces much worse results than our method. This experiment proves that the increased amount of concepts in larger databases is not the main cause of our performance improvements. In fact, it seems that such information cannot be trivially harnessed, and our method is needed to enable information extraction from these large databases. As demonstrated by our ablations (Tab. 4a), it is our specific design choices (e.g., the multimodal candidate scoring) that contribute to the performance improvements.
> - In the preliminary study (see Fig. 2 and subsection “Semantic space representation”), we showed that captions retrieved from databases are better than the generated captions for classification tasks. This serves as the basis for our method design. It is also the main contributor to the improvement in performance w.r.t. BLIP-2 VQA. In fact, by having a decoder, BLIP-2 is not guaranteed to always output meaningful information, thus hindering the classification performance. For instance, in some cases we found that the list of candidates generated by BLIP-2 contained repeated classes (e.g., ["dog", "dog", "dog"]).
>
>
> **[W3: Speed vs parameters]** We agree with the reviewer that only mentioning the number of parameters in L164-165 is not precise enough and does not directly quantify speed. We will revise the statement accordingly. Actually, we reported the inference time on an NVIDIA A6000 GPU over a batch of size 64 in Tab. 2 of the Supplementary where we observe that retrieval from databases of millions of samples can be extremely efficient. The inference time of our method takes approximately 4370 ms to predict a batch of 64, while the smallest BLIP-2 model (i.e., ViT-L) takes around 5670 ms. Last, we also show that CaSED with our subset of PMD (around 55M captions) or with LAION-400M (around 414M) takes approximately the same inference time, a promising sign for efficient database scaling.
>
>
> **[Q1: Other scoring strategies]** We experimented with the suggested strategy for computing the final candidate scores, by averaging the cosine similarities from the two modalities before applying the softmax transformation. We test this aggregation strategy on the ten datasets used in the main manuscript and report the average, with the relative gain highlighted w.r.t. our approach (averaging probabilities). For cluster accuracy, we achieve 43.0% (-0.1%), for semantic IOU 17.7% (+0.0%), for semantic similarity 50.8 (+0.4). The experiment shows that there is no significant difference between the two approaches.  Instead, our candidate generation and multimodal scoring, as ablated in the main manuscript, are the key components for our advantageous performance. We report the complete results of the suggested ablation in Tab. A1 in the rebuttal PDF.

---

> > ### Comment · Reviewer_panm · 2023-08-20
> >
> > Thank you for addressing my concerns and questions. I have increased my score accordingly.

---

### Author Rebuttal · Authors · 2023-08-09

We thank the reviewers for their valuable feedback. We are especially glad the reviewers appreciated i) the novel task, with a well-thought evaluation protocol and baselines (m2jc, xx1r, 3fhc), ii) the proposed training-free method, which consistently outperforms multiple baselines on several datasets (panm, xx1r), iii) the clear explanations and motivations, with good and interesting ablations to justify our design choices (panm, m2jc, 3fhc).

Here we provide details on the clustering accuracy metric, a shared question among reviewers. Below, we answer specific concerns with comments on the respective reviews. We also attach a PDF file with additional experiments requested by reviewers to further demonstrate the design choices and validate the strengths of our method. In Tab. A1, we compare with new baselines (e.g., larger vocabulary, occurrence-based scoring). In Tab. A2, we report statistics on the distribution of retrieved and selected candidates. Last, in Tab. A3 we expand the database analysis with their relevance and statistics on their composition. From now on, we will refer to the PDF attached to this comment as the "rebuttal PDF".

Regardless of the outcome, we plan to release the complete codebase (metrics, baselines, databases, etc.) to foster further research on the topic.


**[Global response on cluster accuracy]** To evaluate cluster accuracy, we take inspiration from the evaluation protocols used for deep visual clustering [a,b,c]. After grouping the samples by their predicted semantic label, we solve a linear mapping problem to link each predicted cluster to a single ground-truth cluster. This mapping is resolved with a many-to-one match, where a predicted cluster is assigned to the most present ground-truth label. The many-to-one match is justified by the vocabulary-free paradigm. It could happen that a vocabulary-free method generates different clusters for the same ground-truth label simply due to, e.g., synonyms. For instance, a vocabulary-free method could assign labels “insect” and “bug” for images labeled as “insects”.

Of course, this is prone to degenerate cases. For instance, if all samples are assigned to a unique predicted cluster, the cluster accuracy equals the ratio of the most present class over the dataset size. On the other hand, if we have a number of clusters equal to the number of samples, we would achieve 100% accuracy. However, these phenomena never happen in the methods we test, whether we generate captions (i.e., with BLIP-2), we retrieve from text databases (i.e., Knowledge Base), or we use fixed vocabularies (simple baselines). More precisely, here are the average numbers of predicted clusters:
- CaSED: ~800 clusters
- BLIP-2 (VQA): ~900 clusters
- BLIP-2 (caption): ~3000 clusters
- CLIP + WordNet: ~1300 clusters
- CLIP + English Words: ~1600 clusters
- CLIP + PMD Words: ~3900 clusters

All the methods produce a comparable number of clusters. While having more clusters might improve performance according to how the metric is computed, this does not happen in practice. We believe that selecting good candidates is ultimately more important than having many candidates, as it enables better grouping, increasing clustering performance.

Finally, we would like to highlight that clustering accuracy is only one of the metrics that we exploit. Together with the other metrics, our evaluation paints a clear picture:
- Cluster accuracy evaluates how well predictions are grouped together and consistent
- Semantic similarity assesses how semantically close the predicted and ground truth class are, with quasi-invariance to synonyms and orthographic differences
- Semantic IoU measures how frequently methods output the exact class name or a subset of it

Despite these differences, the ranking of the methods in our experiments is roughly the same for all the metrics, and ours consistently outperforms the others.

[a] Van Gansbeke, et al. "Scan: Learning to classify images without labels" ECCV 2020.\
[b] Ji, Henriques, and Vedaldi. "Invariant information clustering for unsupervised image classification and segmentation" ICCV 2019.\
[c] Han, et al. "Automatically Discovering and Learning New Visual Categories with Ranking Statistics" ICLR 2020.

---

### Decision · Program_Chairs · 2023-09-21

**Decision:**

Accept (poster)

**Comment:**

After the rebuttal and discussion, the majority of reviewers recommend acceptance. There is a remaining concern of reviewer xx1r about contribution and novelty. The AC read the paper, reviews and discussion and also all related work mentioned in the discussion and finds no strong concerns about the contribution or novelty of this submission, especially when disregarding contemporaneous work. The paper proposes lifting the restriction of zero-shot classification to a pre-defined set of classes. As noted by the reviews this might not be fully well-defined as depending on the granularity of the target task the overall vocabulary might still need to change. Additionally, images can be classified into multiple classes which is difficult to model. That said, the task is interesting and the method, although somewhat simple improves over the necessary baselines by a good margin. The field of open-vocabulary classification will benefit from this paper and it should be accepted. Please make sure to update the paper with all the promised changes, discussions, and references for the final version.